# Protective effects of combining monoclonal antibodies and vaccines against the *Plasmodium falciparum* circumsporozoite protein

**Lawrence T. Wang**[1], **Lais S. Pereira**[1], **Patience K. Kiyuka**[1,2], **Arne Schön**[3], **Neville K. Kisalu**[1], **Rachel Vistein**[1], **Marlon Dillon**[1], **Brian G. Bonilla**[1], **Alvaro Molina-Cruz**[4], **Carolina Barillas-Mury**[4], **Joshua Tan**[5], **Azza H. Idris**[1,6], **Joseph R. Francica**[1], **Robert A. Seder**[1] *

**1** Vaccine Research Center, National Institute of Allergy and Infectious Diseases, National Institutes of Health, Bethesda, Maryland, United States of America, **2** Department of Biological Sciences, Pwani University, Kilifi, Kenya, **3** Department of Biology, Johns Hopkins University, Baltimore, Maryland, United States of America, **4** Laboratory of Malaria and Vector Research, National Institute of Allergy and Infectious Diseases, National Institutes of Health, Rockville, Maryland, United States of America, **5** Antibody Biology Unit, Laboratory of Immunogenetics, National Institute of Allergy and Infectious Diseases, National Institutes of Health, Rockville, Maryland, United States of America, **6** The Ragon Institute of Massachusetts General Hospital, Massachusetts Institute of Technology and Harvard University, Cambridge, Massachusetts, United States of America

* rseder@mail.nih.gov

**Data Availability Statement:** All relevant data are within the manuscript and its Supporting Information files.

## Abstract

Combinations of monoclonal antibodies (mAbs) against different epitopes on the same antigen synergistically neutralize many viruses. However, there are limited studies assessing whether combining human mAbs against distinct regions of the *Plasmodium falciparum* (Pf) circumsporozoite protein (CSP) enhances *in vivo* protection against malaria compared to each mAb alone or whether passive transfer of PfCSP mAbs would improve protection following vaccination against PfCSP. Here, we isolated a panel of human mAbs against the subdominant C-terminal domain of PfCSP (C-CSP) from a volunteer immunized with radiation-attenuated Pf sporozoites. These C-CSP-specific mAbs had limited binding to sporozoites *in vitro* that was increased by combination with neutralizing human "repeat" mAbs against the NPDP/NVDP/NANP tetrapeptides in the central repeat region of PfCSP. Nevertheless, passive transfer of repeat- and C-CSP-specific mAb combinations did not provide enhanced protection against *in vivo* sporozoite challenge compared to repeat mAbs alone. Furthermore, combining potent repeat-specific mAbs (CIS43, L9, and 317) that respectively target the three tetrapeptides (NPDP/NVDP/NANP) did not provide additional protection against *in vivo* sporozoite challenge. However, administration of either CIS43, L9, or 317 (but not C-CSP-specific mAbs) to mice that had been immunized with R21, a PfCSP-based virus-like particle vaccine that induces polyclonal antibodies against the repeat region and C-CSP, provided enhanced protection against sporozoite challenge when compared to vaccine or mAbs alone. Collectively, this study shows that while combining mAbs against the repeat and C-terminal regions of PfCSP provide no additional protection *in vivo*, repeat

**Funding:** L.T.W., L.S.P., P.K.K., N.K.K., R.V., M.D., B.G.B., A.M., C.B., J.T., A.H.I., J.R.F., and R.A.S. were supported by the Intramural Research Division of the National Institute of Allergy and Infectious Diseases; A.S. was supported by the National Cancer Institute (HHSN261200800001E). The funders had no role in study design, data collection and analysis, decision to publish, or preparation of the manuscript.

**Competing interests:** I have read the journal's policy and the authors of this manuscript have the following competing interests: R.A.S., J.F., L.T.W., and R.V. have submitted U.S. Provisional Patent Application No. 62/842,590, filed 3 May 2019, describing mAb L9. All other authors have declared that no competing interests exist.

mAbs do provide increased protection when combined with vaccine-induced polyclonal antibodies. These data should inform the implementation of PfCSP human mAbs alone or following vaccination to prevent malaria infection.

## Author summary

The *Plasmodium falciparum* (Pf) circumsporozoite protein (CSP) is the major surface protein on sporozoites and is required for these malaria parasites to invade the liver. Antibodies can prevent malaria by neutralizing sporozoites prior to liver invasion. The only approved malaria vaccine (RTS,S) is comprised of the repeat region and C-terminus of PfCSP. RTS,S-mediated protection is associated with vaccine-induced antibodies against both regions. While monoclonal antibodies (mAbs) against the three tetrapeptides in the repeat region potently bind and neutralize sporozoites, mAbs against the C-terminus demonstrate limited sporozoite binding and neutralization. Here, we used flow cytometry to show that the sporozoite binding of C-terminal mAbs are potentiated by combining them with repeat mAbs. This *in vitro* synergy did not translate into enhanced *in vivo* protection against sporozoite challenge in mice treated with repeat and C-terminal mAb combinations. Furthermore, combining mAbs against the three tetrapeptides in the repeat region did not provide enhanced protection against sporozoite challenge compared to each mAb alone. However, combining passive and active immunization with repeat mAbs and a RTS,S-like vaccine improved protection against sporozoite challenge compared to each intervention alone. These results have important implications for implementing anti-PfCSP mAbs alone or in combination with vaccines to prevent malaria.

## Introduction

Malaria is a parasitic mosquito-borne disease that affected 200–400 million people and led to ~400,000 deaths in 2019, mostly from *Plasmodium falciparum* (Pf) infection [1]. Antibodies can prevent malaria by inhibiting sporozoites (SPZ; the infectious form of *Plasmodium* parasites transmitted by mosquitos) before they invade hepatocytes in the liver. Antibody-mediated SPZ inhibition can be achieved by direct neutralization (e.g., blocking parasite motility [2] or invasion of hepatocytes [3,4]) or by engaging Fc-mediated effector functions such as opsonophagocytosis [5] or complement fixation [6]. Most anti-SPZ antibodies induced following natural infection or immunization with attenuated SPZ target the circumsporozoite protein (CSP), the most abundant surface protein on SPZ [7]. CSP has three domains: an N-terminus (N-CSP), a central region composed of repeating tetrapeptides (1 NPDP, 4 NVDP, and 38 NANP in the Pf reference strain 3D7), and a C-terminus (C-CSP) containing a linker sequence, an α-thrombospondin type-1 repeat domain (αTSR), and a glycosylphosphatidylinositol (GPI) anchor sequence (Fig 1) [8].

RTS,S, the only malaria vaccine that has been recommended for usage by the World Health Organization [9], is a truncated form of PfCSP_3D7 containing 19 NANP repeats and C-CSP minus the GPI anchor [10]. RTS,S induces antibodies against the repeat region and C-CSP [11–13], and antibodies against the immunodominant NANP repeats correlate with RTS,S-mediated protection in African infants and young children [14]. Systems serology analysis of RTS,S vaccinees found that opsonophagocytosis and engagement of Fc gamma receptor 3A were most predictive of RTS,S protection [15]. Additionally, protective CSP antibodies

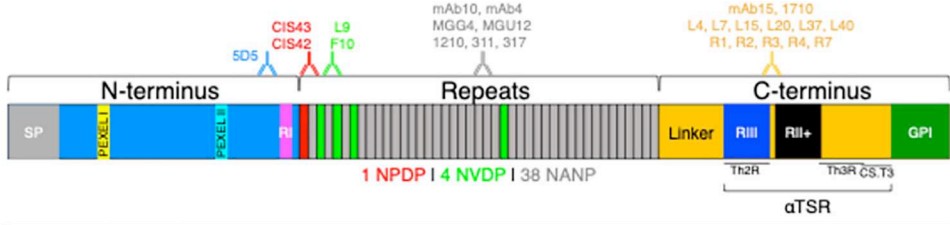

SP, signal peptide; PEXEL, Plasmodium export element; R, region; GPI, glycosylphosphatidylinositol anchor; Th2R/Th3R/CS.T3, CD4 helper T cell epitopes; α-TSR, α1-helix thrombospondin type-1 repeat

**Fig 1. Schematic/sequence of PfCSP_3D7 and approximate epitopes bound by mAbs used in this study.** Top: color-coded schematic illustrating the N-terminus (N-CSP), repeat region, and C-terminus (C-CSP) of PfCSP_3D7. N-CSP contains a signal peptide (SP), two Plasmodium export element (PEXEL) sites, and the conserved region I (RI). The repeat region is composed of three types of tetrapeptides (1 NPDP, 4 NVDP, and 38 NANP). C-CSP has a linker to the repeat region, an α-thrombospondin type-1 repeat (αTSR) domain that contains two conserved motifs (region III and region II+, RIII and RII+) and several CD4+ helper T cell epitopes (Th2R, Th3R, CS.T3), and a glycosylphosphatidylinositol (GPI) anchor sequence. The binding sites of monoclonal antibodies used in this study are depicted. Bottom: sequence of PfCSP_3D7, color-coded to match the schematic. The sequence of the recombinantly expressed C-CSP construct used in this study is underlined.

induced in children after natural malaria infection [6] or immunization of pre-exposed adults with whole attenuated sporozoite [16] were shown to activate complement *in vitro*. Together, these data suggest that PfCSP antibodies utilize multiple mechanisms to prevent SPZ from invading hepatocytes.

The majority of anti-PfCSP monoclonal antibodies (mAbs) isolated to date are NANP-specific [17], with a minority being shown to potently protect mice from challenge with transgenic *P. berghei* SPZ expressing full-length PfCSP [18–21]. However, there is some evidence that C-CSP antibodies contribute to SPZ neutralization. Specifically, serum from mice immunized with C-CSP peptides inhibits SPZ hepatocyte invasion *in vitro* [22,23]. In the context of RTS, S-immunized adults and/or children, C-CSP antibodies in polyclonal serum mediate phagocytosis and complement fixation *in vitro* [24,25] and C-CSP-specific IgG avidity and breadth correlate with RTS,S efficacy [26,27]. C-CSP IgG$_2$/IgA$_2$ [15] and C-CSP IgG$_4$ [28] have been associated with protection; however, another report showed that C-CSP IgG$_{2/4}$ were associated with increased malaria risk whereas C-CSP IgG$_{1/3}$ were associated with protection [29]. Moreover, genetic analyses of Pf field isolates showed that C-CSP is highly polymorphic and that RTS,S is less effective against non-3D7 strains, suggesting that C-CSP is under substantial immune pressure [30,31]. Together, these data suggest that C-CSP antibodies neutralize SPZ *in vitro* and may be associated with RTS-S-mediated protection *in vivo* when accompanied by repeat antibodies.

Based on these data, efforts have been made to isolate and characterize mAbs targeting C-CSP from malaria-naïve PfSPZ-immunized adult volunteers, resulting in the publication of three C-CSP-specific human mAbs to date [21,32]. Despite displaying high-affinity binding to full-length recombinant PfCSP (FL-rCSP), these three C-CSP mAbs had minimal SPZ binding and neutralization *in vitro* and *in vivo* [21,32], suggesting that C-CSP may be inaccessible on

the surface of SPZ. Indeed, native PfCSP on SPZ has been shown to undergo reversible conformational changes to mask C-CSP [22,33], with the repeat region acting as a flexible spring between N- and C-CSP [34,35]. These data indicate that PfCSP is structurally labile and may adopt different conformations to mask C-CSP from antibody recognition.

Conformational masking has been described for viral glycoproteins like gp120 on HIV-1; furthermore, the binding of certain antibodies has been shown to trigger additional conformational changes in gp120 [36]. Similarly, binding of the human PfCSP mAb CIS43 to the unique tetrapeptide NPDP at the junction of N-CSP and the repeat region has been proposed to induce conformational changes in FL-rCSP [21]. These data suggest that combining a repeat mAb that changes the conformation of PfCSP with a conformation-dependent C-CSP mAb may improve the binding and neutralization of the C-CSP mAb. Indeed, combining mAbs that bind different epitopes on gp120 has been shown to synergistically block HIV-1 infection [37,38].

Here, to further define the role of C-CSP-specific antibodies in preventing malaria infection and investigate whether combining mAbs targeting different PfCSP epitopes results in improved protection against SPZ challenge, we isolated a large panel of human C-CSP mAbs and characterized their *in vitro* SPZ binding and *in vivo* SPZ neutralization alone or in combination with neutralizing repeat mAbs. We extended this analysis by assessing whether combining potent NPDP/NVDP/NANP-specific repeat mAbs provided increased protection *in vivo* compared to single repeat mAbs. Finally, we evaluated whether passive administration of repeat- or C-CSP-specific mAbs in mice immunized with a next-generation RTS,S-like vaccine called R21 [39] provided improved protection compared to each intervention alone. Collectively, these data provide insight into how PfCSP mAbs can be optimally used to prevent malaria.

## Results

### C-CSP-specific mAbs do not bind or neutralize SPZ

To assess the ability of C-CSP-specific mAbs to bind and neutralize SPZ, a panel of thirteen C-CSP-specific human mAbs was assembled (Fig 1 and S1 and S2 Tables). Two mAbs in the panel, 1710 and mAb15, have been previously reported [21,32]. The panel's eleven other C-CSP mAbs were isolated from a PfSPZ-immunized subject [40]. Consistent with previous reports on human PfCSP mAbs isolated from PfSPZ-immunized subjects [19–21,40,41], sequence analysis of the C-CSP mAbs showed that they had low somatic mutations (0.69–5.5% in the heavy chain and 0.35–3.23% in the light chain; S2 Table). Notably, most (11 of 13) C-CSP mAbs had lambda light chains (S1A Fig and S2 Table).

All C-CSP mAbs in the panel bound FL-rCSP and C-CSP, but not N-CSP or the 36mer repeat peptide (NANP)$_9$, by ELISA (Fig 2A). Notably, C-CSP was expressed in mammalian cells and only includes part of the linker sequence and the entire αTSR subdomain (Fig 1). To determine the fine epitopes recognized by these mAbs, mapping was performed using overlapping 15mer linear peptides spanning C-CSP. None of the C-CSP mAbs had detectable peptide binding by ELISA (S1B Fig), corroborating that C-CSP mAbs recognize conformational C-CSP epitopes [32]. To extend this analysis, we used surface plasmon resonance (SPR) to perform epitope binning of the 13 C-CSP mAbs binding to FL-rCSP, with the N-CSP mAb 5D5 and the NANP-specific mAb10 included as controls (Fig 2B and S3 Table). The C-CSP mAbs fell broadly into three bins. Most mapped to the same bin as 1710, which binds an epitope overlapping the Th2R/Th3R T cell epitopes in the αTSR (Fig 1) and is the only C-CSP mAb whose structure has been solved [32]. Notably, R2/R7 and L4 mapped to different bins.

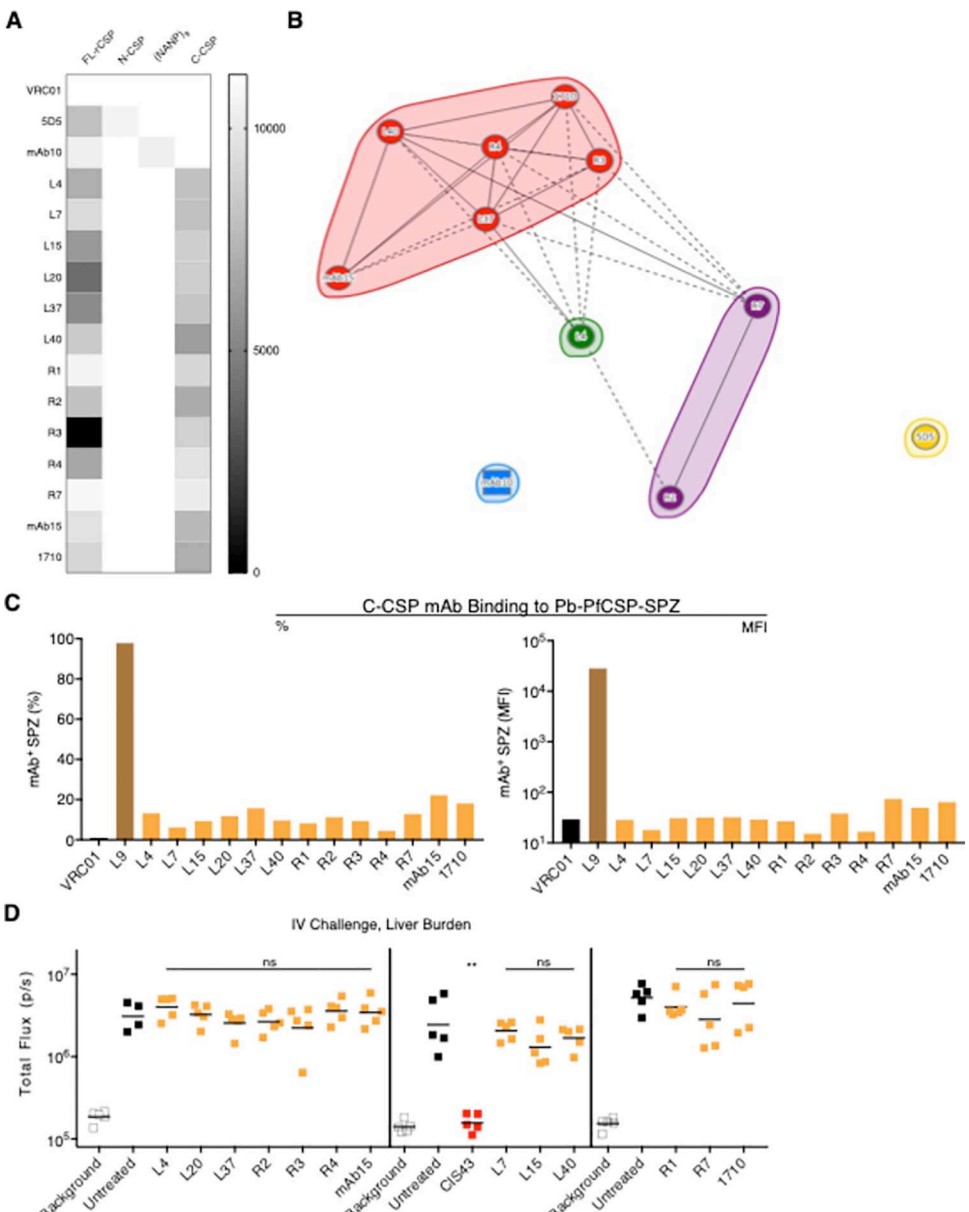

**Fig 2. Binding and *in vivo* protection mediated by C-CSP-specific mAbs. (A)** Heat map of area under the curve (AUC) values of thirteen C-CSP mAbs binding to FL-rCSP, N-CSP, 36mer peptide (NANP)$_9$, and C-CSP by ELISA. VRC01 (anti-gp120 mAb), 5D5 (N-CSP mAb), and mAb10 (NANP-preferring repeat mAb) were respectively included as a negative isotype control, positive control N-CSP mAb, and positive control repeat mAb. Data were averaged from 2–3 independent experiments. **(B)** Epitope binning of C-CSP mAbs binding to FL-rCSP measured by SPR. All mAbs were tested as both ligands and analytes; several mAbs (L7, L15, L20, R1, mAb10) were excluded due to poor ligand and/or analyte binding to FL-rCSP. Solid lines indicate two-way competition; dotted lines indicate one-way competition. **(C)** Percentage (%) and median fluorescence intensity (MFI) of Pb-PfCSP-SPZ bound by 20 µg/mL of indicated mAb, measured by flow cytometry. VRC01 and L9 (NVDP-preferring repeat mAb) were included as negative and positive controls. **(D)** Liver burden (bioluminescence; total flux, photons/sec) in mice 40 hours post-challenge (n = 5/group; line indicates geometric mean) mediated by 300 µg of indicated mAbs administered 2 hours before IV challenge with 2,000 Pb-PfCSP-SPZ. CIS43 (NPDP-preferring repeat mAb) was included as a positive control. Vertical lines separate independent experiments. *P*-values were determined by comparing mAbs to untreated control using the Kruskal-Wallis test with Dunn's post-hoc correction. **, $p < 0.01$; ns (not significant), $p > 0.05$.

Together, these data indicate that most of the C-CSP mAbs compete for the same epitope in the αTSR of the C-CSP.

However, all C-CSP mAbs showed low *in vitro* binding to transgenic *P. berghei* SPZ expressing full-length PfCSP and a green fluorescent protein/luciferase fusion protein (Pb-PfCSP-GFP/Luc-SPZ; hereafter Pb-PfCSP-SPZ) harvested from the salivary glands of mosquitos (Fig 2C). Moreover, passive transfer of 300 μg of each C-CSP mAb into mice did not lower parasite liver burden following intravenous (IV) challenge with Pb-PfCSP-SPZ (Fig 2D). As a positive control, 300 μg/mouse of the potent NPDP-specific repeat mAb CIS43 (S1 Table) lowered liver burden to background levels and completely protected mice from SPZ challenge. These data show that, despite binding recombinant PfCSP, C-CSP-specific mAbs do not bind native PfCSP on salivary gland SPZ and are non-neutralizing *in vivo*.

## C-CSP mAbs can distinguish between different PfCSP conformations on SPZ

Prior studies have suggested that CSP on SPZ isolated from mosquito salivary glands (SG) and midguts (MG) adopt two conformations, with C-CSP being masked ("closed conformation") on SG-SPZ and exposed ("open" conformation) on MG-SPZ [33]. We hypothesized that C-CSP mAbs might differentiate between the "closed" and "open" conformations of PfCSP; thus, we compared the binding of several C-CSP mAbs to SG- and MG-SPZ isolated from Pb-PfCSP-SPZ-infected mosquitos. Consistent with Fig 2C, all C-CSP mAbs showed low binding to "closed" PfCSP on SG-SPZ (10–20% mAb$^+$ SPZ) and increased binding to "open" PfCSP on MG-SPZ (20–88% mAb$^+$ SPZ), most notably mAb15 and L15 (Fig 3A and 3B). These data further substantiate the existence of distinct PfCSP conformations on MG- versus SG-SPZ [33] and suggest that C-CSP mAbs can be used to distinguish between native PfCSP conformations.

## Repeat mAbs can change the conformation of recombinant PfCSP

We previously reported that mAb CIS43 binds to FL-rCSP with two affinities in a sequential fashion. Furthermore, we suggested it was possible that the first high-affinity binding event caused conformational changes in FL-rCSP that enabled a second, lower affinity binding event to several other sites on FL-rCSP [21]. In a subsequent study, we showed that mAbs L9, 311, and 317 also exhibited a similar "two-step binding" phenotype to FL-rCSP [40]. Conformational changes should be accompanied by significant changes in enthalpy (ΔH) due to burial of hydrophobic amino acid residues away from water, which manifest as large and negative changes in heat capacity (ΔC$_p$) [42,43]. Thus, to assess if the binding of these four two-step binding mAbs induce conformational changes in FL-rCSP, ΔC$_p$ for the binding of each mAb was determined by measuring ΔH at different temperatures (S2A Fig).

For the first binding step, large negative ΔC$_p$ values were observed for 311 and CIS43 (ΔC$_{p1}$ = -1.75 and -0.89 kcal/K/mol, respectively) while the values for L9 and 317 (ΔC$_{p1}$ = -0.27 and -0.35 kcal/K/mol, respectively) were considerably smaller and closer to the magnitudes expected for binding events where only residues in the interacting surfaces contribute to the binding energetics (S2B Fig). The heat capacity changes associated with the second binding events (ΔC$_{p2}$) were modest for all four mAbs (ΔC$_{p2}$ range = -0.02 ➔ -0.25 kcal/K/mol). Taken together, these results suggest that, of the four two-step binding mAbs, only 311 and CIS43 induced significant conformational changes in FL-rCSP and only during the first binding step. These data are consistent with previous reports showing that 311, but not 317, Fabs induce a conformational change in the NANP repeats that cause rCSP to adopt an extended spiral conformation [44,45].

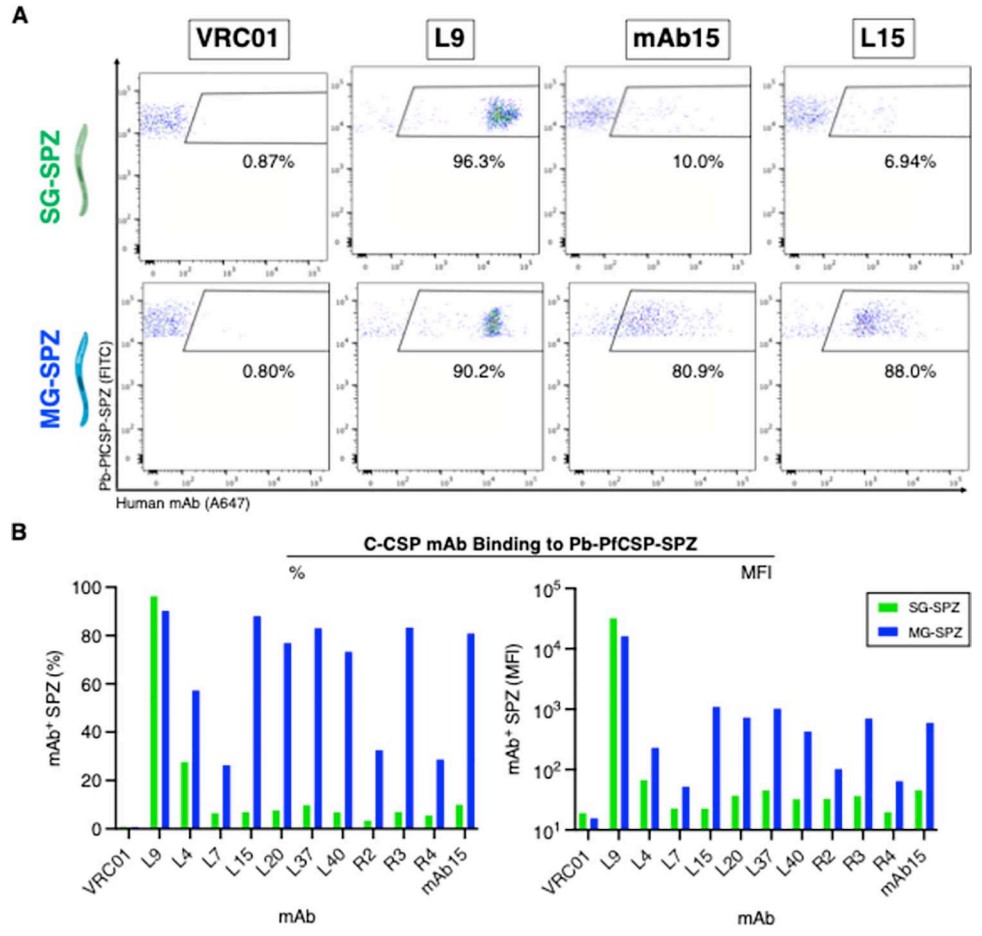

**Fig 3. C-CSP mAbs differentiate between PfCSP conformations on Pb-PfCSP-SPZ. (A)** Representative flow cytometry plots depicting 20 μg/mL of indicated mAb binding to Pb-PfCSP-SPZ harvested from the salivary glands (SG-SPZ, green) or midguts (MG-SPZ, blue) of mosquitos. **(B)** Percentage and MFI of SG vs. MG Pb-PfCSP-SPZ bound by 20 μg/mL of indicated mAb, measured by flow cytometry. VRC01 and L9 (NVDP-specific repeat mAb) were included as negative and positive controls.

## Repeat mAbs potentiate C-CSP mAb SPZ binding by changing the conformation of PfCSP

Having observed that certain repeat mAbs (311 and CIS43) can change the conformation of FL-rCSP, we hypothesized that the binding of repeat mAbs to SG-SPZ might change the conformation of native PfCSP and improve C-CSP mAb binding. To test this, Alexa Fluor 750-labeled (A750) mAb15 and L15, the two C-CSP mAbs that most clearly distinguished between the "open" vs. "closed" conformations of PfCSP on MG- vs. SG-SPZ (Fig 3B), were used as reporters for the conformation of native PfCSP. SG Pb-PfCSP-SPZ or PfSPZ were co-incubated with unlabeled NPDP-, NVDP-, or NANP-preferring repeat mAbs of various potencies (S1 Table) and either mAb15- or L15-A750 before flow cytometry analysis (Figs 4A and 4B and S3A).

Remarkably, all nine repeat mAbs tested (including 311 and CIS43) comparably improved mAb15- and L15-A750 SG-SPZ binding from 10–20% to 80–90% mAb+ SPZ, with a >100-fold increase in median fluorescence intensity (MFI) (Figs 4C and S3B). The potentiation of C-CSP mAb binding was specific to repeat mAbs, as co-incubating mAb15- and

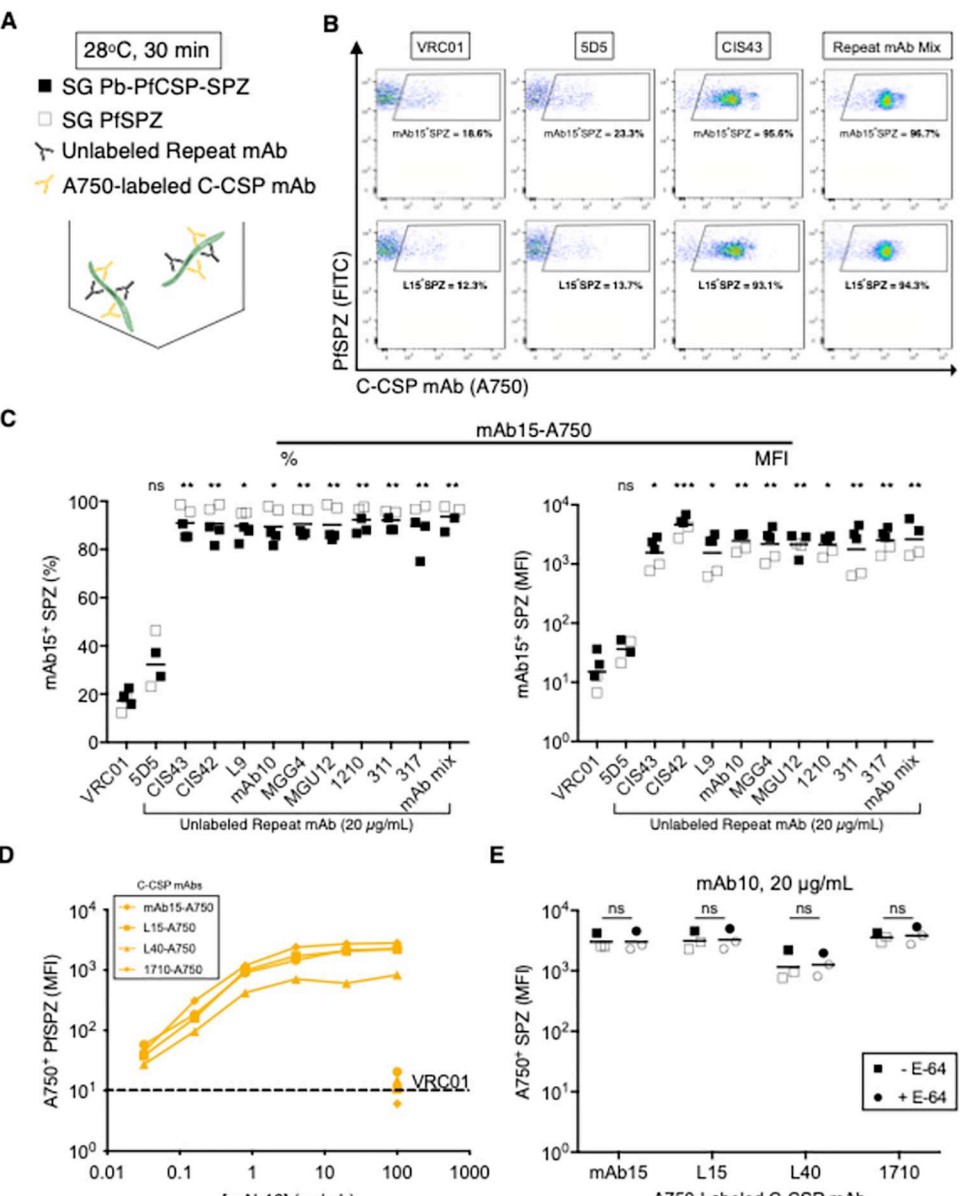

**Fig 4. PfCSP repeat mAbs potentiate the SPZ binding of C-CSP mAbs. (A)** Experimental schema for measuring the binding of Alexa Fluor 750-labeled mAbs to SG Pb-PfCSP-SPZ or PfSPZ in the presence of unlabeled mAbs. **(B)** Representative flow cytometry plots depicting 20 μg/mL mAb15-A750 (top panel) or L15-A750 (bottom panel) binding to SG PfSPZ in the presence of 20 μg/mL unlabeled VRC01, 5D5, CIS43, and repeat mAb mix (mAb mix, CIS43-317 in C). Percentages of mAb15+ or L15+SPZ are shown. **(C)** Percentage and MFI of Pb-PfCSP-SPZ (filled squares) and PfSPZ (open squares) bound by 20 μg/mL mAb15-A750 when co-incubated with 20 μg/mL of specified unlabeled mAb. *P*-values were determined by comparing PfCSP mAbs to VRC01 using the Kruskal-Wallis test. **(D)** Dose-dependent improvement in PfSPZ binding of four A750-labeled C-CSP mAbs (20 μg/mL) mediated by increasing concentrations (0.032–100 μg/mL) of unlabeled NANP-preferring mAb10; 100 μg/mL unlabeled VRC01 was set as the baseline (dotted line). Data are representative of two independent experiments. **(E)** Binding of four A750-labeled C-CSP mAbs (20 μg/mL) to Pb-PfCSP-SPZ (filled symbols) or PfSPZ (open symbols) when co-incubated with unlabeled mAb10 (20 μg/mL), with or without the protease inhibitor E-64 (squares, no E-64; circles, + E-64). *P*-values were determined by comparing -E-64 to +E-64 for each C-CSP mAb using a two-way ANOVA with Sidak's post-hoc correction. (C, E): ***, $p<0.001$; **, $p<0.01$; *, $p<0.05$; ns (not significant), $p>0.05$.

L15-A750 with the N-CSP mAb 5D5 did not alter their binding. Furthermore, mixing all the repeat mAbs in the panel (Repeat mAb Mix) to simulate a polyclonal response improved mAb15- and L15-A750 binding comparably to each repeat mAb alone. PfCSP repeat mAb-mediated potentiation of C-CSP mAb SPZ binding was dose-dependent, as >4 µg/mL of the NANP-preferring mAb10 was required to maximally improve C-CSP mAb binding (Fig 4D).

It has been proposed that N-CSP masks C-CSP on SG-SPZ and that cleavage of N-CSP reveals C-CSP [22]. Thus, we tested whether E-64, a cysteine protease inhibitor that inhibits N-CSP cleavage [46], affected C-CSP mAb SPZ binding. Adding E-64 did not significantly alter C-CSP mAb binding to either Pb-PfCSP-SPZ or PfSPZ when co-incubated with unlabeled VRC01 or mAb10, suggesting that cleavage of N-CSP does not affect C-CSP accessibility to antibodies (Figs 4E and S3C). Collectively, these data show that the binding of antibodies to the flexible repeat region potentiates the SG-SPZ binding of C-CSP-specific mAbs, likely by changing the conformation of native PfCSP to unmask C-CSP.

## Repeat and C-CSP mAb combinations do not cooperatively neutralize SPZ *in vivo*

The demonstration that repeat mAbs improve the SPZ binding of C-CSP mAbs *in vitro* (Figs 4 and S3) raised the possibility that repeat mAbs could act cooperatively with C-CSP mAbs to neutralize SPZ *in vivo*. To assess this, all C-CSP mAbs were combined with mAb10, a NANP-preferring repeat mAb that binds FL-rCSP in a single step [21] and improves C-CSP mAb SPZ binding comparably to all repeat mAbs tested (Fig 4C). Given the dose-dependent potentiation of C-CSP mAb SPZ binding by repeat mAbs (Fig 4D), we reasoned that selecting mAb10, which is not the most protective repeat mAb in the panel (S1 Table), would allow for higher repeat mAb dosing in mice and thus might increase the chances of detecting *in vivo* cooperativity with C-CSP mAbs.

C-CSP mAb and mAb10 combinations were passively transferred into mice prior to IV challenge with Pb-PfCSP-SPZ. C-CSP mAbs were dosed at 300 µg/mouse while mAb10 was given at 100 µg/mouse, a dose that lowers liver burden to the midpoint of the assay's dynamic range and thus enables detection of any additional protective effects attributable to each C-CSP mAb. The anti-gp120 human mAb VRC01 was included as an isotype control at 300 µg/mouse. None of the IV challenged mice treated with combinations of C-CSP mAbs and mAb10 had liver burdens that were significantly different from mice treated with mAb10 + VRC01 (Fig 5A).

Since the skin is a major physiological site for CSP antibodies to neutralize SPZ [47,48], mice were then challenged by the intradermal (ID) route. For the ID challenge, C-CSP mAbs were again dosed at 300 µg/mouse while the dose of mAb10 was lowered to 30–50 µg/mouse to account for the lower antibody threshold required to protect against ID challenge [40]. As in the IV challenge data, when compared to mAb10 + VRC01 all C-CSP mAb + mAb10 combinations did not significantly lower liver burden or subsequent parasitemia following ID challenge (Fig 5B).

Given the dependence of C-CSP mAbs on the conformation of PfCSP, we expanded the *in vivo* neutralization analysis to include the two-step binding mAbs CIS43 and 311, which both significantly changed the conformation of FL-rCSP by ITC (S2 Fig) and are more potent than mAb10 at lowering liver burden following IV challenge [40]. Combinations of CIS43 or 311 (50 µg/mouse) and mAb15 or L15 (300 µg/mouse) were passively transferred into mice prior to IV challenge. Liver burdens in mice treated with CIS43 or 311 + mAb15 or L15 combinations were similar to mice treated with CIS43 + VRC01 or 311 + VRC01 (S4A Fig). Collectively, these data show that the potentiation of C-CSP mAb SPZ binding by repeat mAbs *in vitro* did not translate into increased SPZ neutralization *in vivo*.

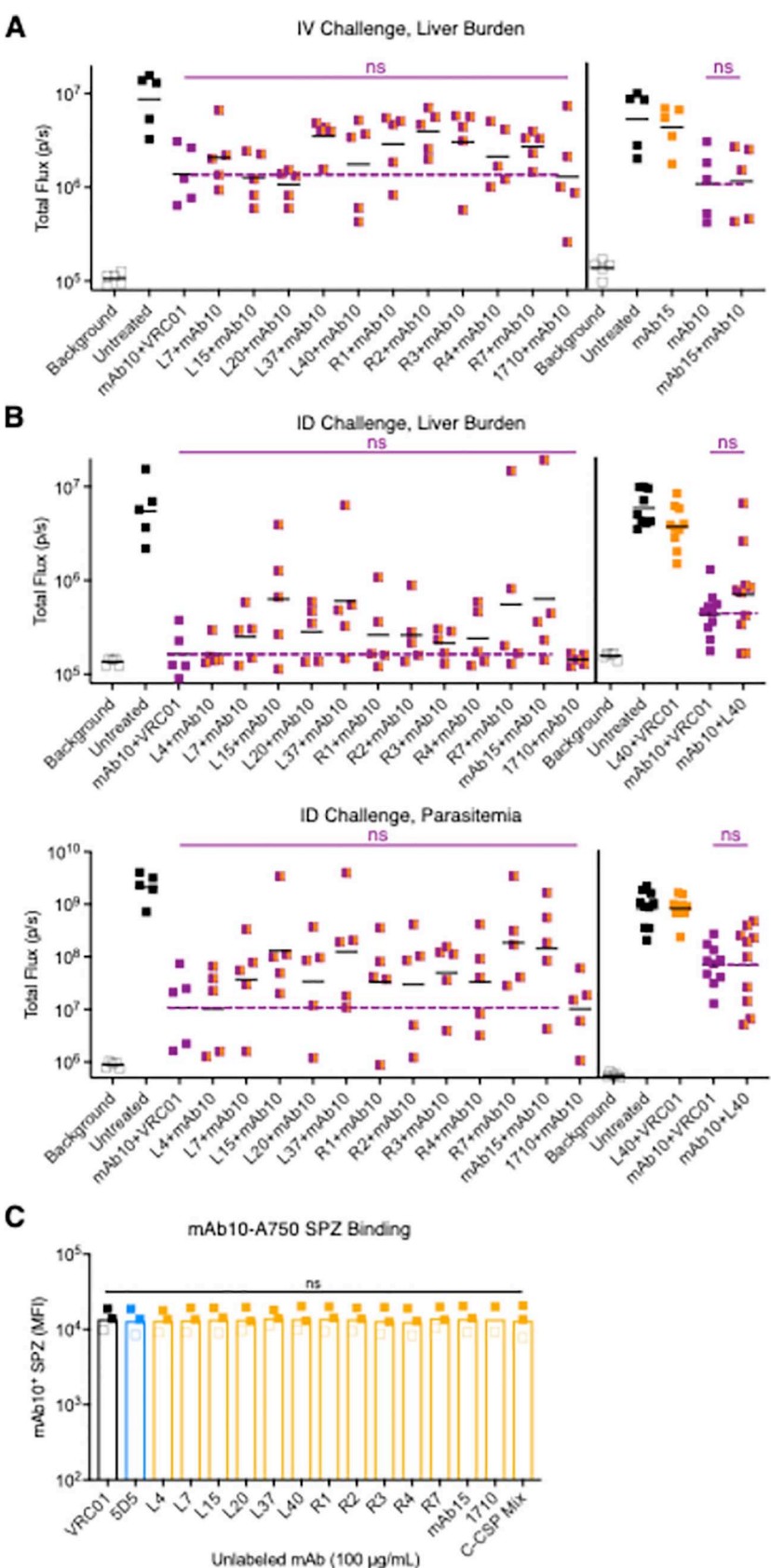

Fig 5. Repeat mAbs do not potentiate the SPZ neutralization of C-CSP mAbs *in vivo*. (A) Liver burden in mice (n = 5/group; line indicates geometric mean) 40 hours post-challenge mediated by indicated mAb combinations (mAb10, 100 μg; C-CSP mAbs or VRC01, 300 μg) administered 2 hours before IV challenge with 2,000 Pb-PfCSP-SPZ. (B) Liver burden (top) and parasitemia (bottom) in mice (n = 5-10/group) respectively 40 hours and 5 days post-challenge mediated by indicated mAb combinations (mAb10, 50 μg in left experiment, 30 μg in right experiment; C-CSP mAbs or VRC01, 300 μg) administered 24 hours before ID challenge with 5,000 Pb-PfCSP-SPZ. (A-B): *P*-values were determined by comparing C-CSP mAb + mAb10 (orange/purple squares) to mAb10 + VRC01 (purple squares) using either the two-tailed Mann-Whitney test (2 comparisons) or the Kruskal-Wallis test with Dunn's post-hoc correction (>2 comparisons). Purple dotted lines were set at mAb10 + VRC01; vertical lines separate independent experiments. (C) MFI of Pb-PfCSP-SPZ (filled squares) and PfSPZ (open squares) bound by 2 μg/mL mAb10-A750 when co-incubated with 100 μg/mL of specified unlabeled mAb. *P*-values were determined by comparing PfCSP mAbs to VRC01 using the Kruskal-Wallis test. (A-C): ***, p<0.001; **, p<0.01; *, p<0.05; ns (not significant), p>0.05.

To determine if the SPZ binding of repeat mAbs was affected by the presence of mAbs targeting N- and C-CSP, Pb-PfCSP-SPZ or PfSPZ were co-incubated with 2 μg/mL A750-labeled mAb10 or CIS43 and 100 μg/mL unlabeled C-CSP mAbs, 5D5, or VRC01. When compared to VRC01 + mAb10 or VRC01 + CIS43, none of the C-CSP mAbs in the panel or 5D5 significantly altered the SPZ binding of mAb10-A750 (Fig 5C) or CIS43–A750 (S4B Fig). These data suggest that the binding of mAbs to N- and C-CSP do not alter the SPZ binding of repeat mAbs *in vitro*.

## Repeat mAb combinations do not cooperatively neutralize SPZ and compete for SPZ binding

Having determined that C-CSP mAbs do not neutralize Pb-PfCSP-SPZ *in vivo* alone or in combination with repeat mAbs, we next focused on assessing whether combining protective mAbs targeting the three types of tetrapeptides in the PfCSP repeat region offered improved protection compared to each mAb alone. CIS43, L9, and 317 are highly potent human PfCSP mAbs respectively classified as NPDP-, NVDP-, and NANP-preferring repeat mAbs (Fig 1 and S1 Table) [40]. Furthermore, CIS43 and L9 are currently undergoing clinical development for malaria prophylaxis.

To assess whether combining these potent mAbs cooperatively improved protection against IV challenge, CIS43, L9, and 317 were tested alone at doses of 50 μg/mouse (a dose that results in breakthrough infection) and in combination with VRC01 or each other (25+25 μg/mouse per mAb) (Fig 6A). Liver burdens in mice that received PfCSP mAb combinations were not significantly different than mice that received single mAbs or both respective PfCSP mAb + VRC01 controls. To extend this analysis, we tested CIS43, L9, and 317 combinations across a range of doses (100, 50, and 25 μg/mouse; S5A–S5C Fig). Concordant with Fig 6A, the mAb combinations largely provided protection that was comparable to single mAbs.

Given the proximity of the different tetrapeptides to each other in the PfCSP repeat region and the promiscuous binding of repeat mAbs to peptides composed of all three tetrapeptides [40,49], we hypothesized that CIS43, L9, and 317 might compete with each other to bind PfCSP repeat epitopes on SPZ. To test this hypothesis, A750-labeled L9 (L9-A750) binding to Pb-PfCSP-SPZ was measured in the presence of varying concentrations of unlabeled CIS43, L9, and 317. All three mAbs reduced L9-A750 SPZ binding, with CIS43 showing the greatest degree of antagonism (Fig 6B). Together, these data show that combining highly potent NPDP-, NVDP-, and NANP-preferring repeat mAbs does not provide improved *in vivo* protection compared to each mAb alone and that even mAbs which preferentially recognize distinct tetrapeptides compete for PfCSP repeat epitopes on SPZ.

As combining protective and non-protective mAbs has been shown to synergistically neutralize Pf blood-stage parasites [50], we also assessed whether combining L9 with three poorly

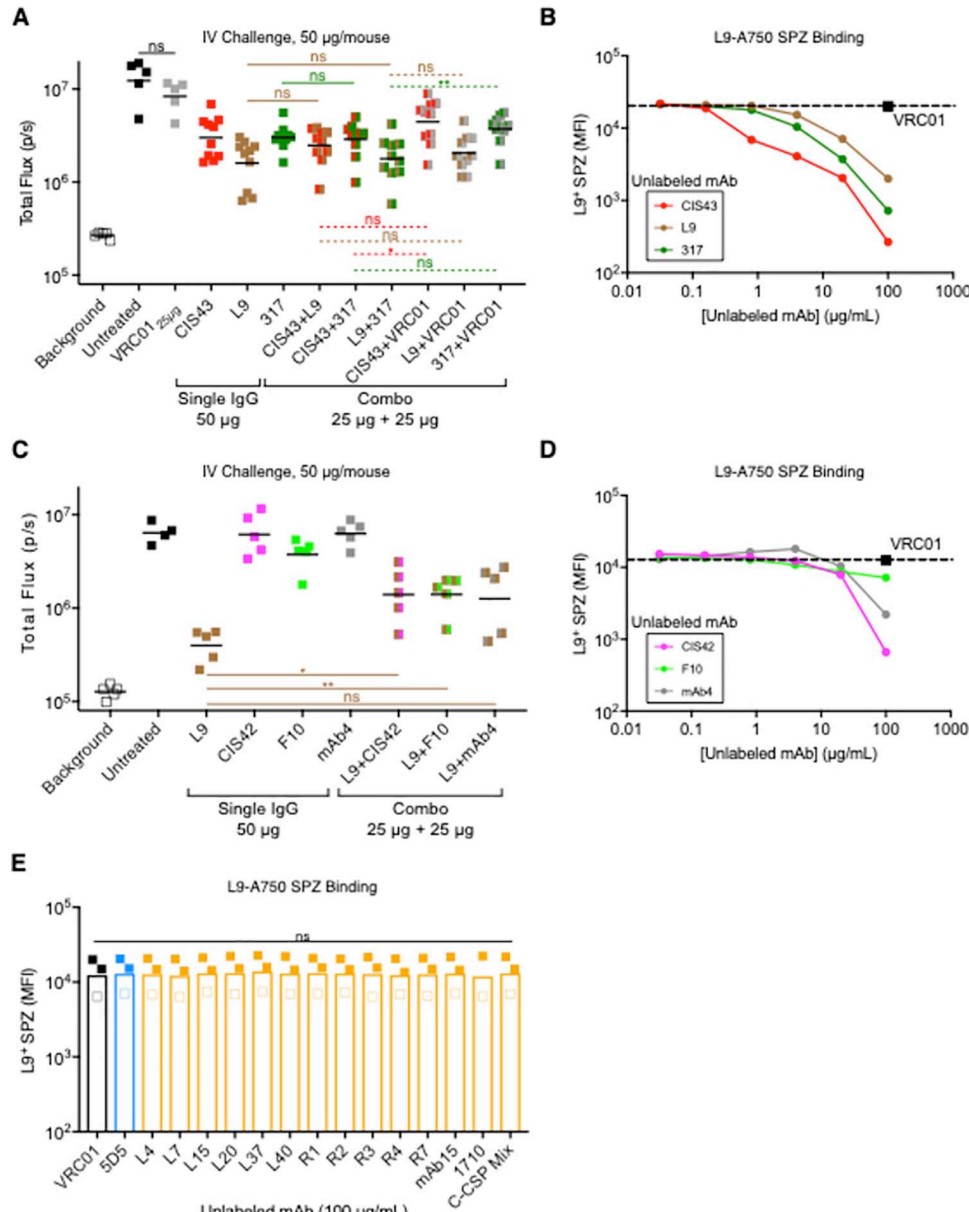

**Fig 6. Repeat mAb combinations do not cooperatively neutralize SPZ *in vivo* and antagonize the SPZ binding of other repeat mAbs *in vitro*. (A)** Liver burden in mice 40 hours post-challenge (n = 10/group) mediated by CIS43, L9, and 317 alone (50 μg) and in combination with each other or isotype control VRC01 (25+25 μg) administered 2 hours before IV challenge with 2,000 Pb-PfCSP-SPZ. *P*-values were determined by comparing PfCSP mAb combinations to L9 or 317 alone using the two-tailed Mann-Whitney test (solid lines) or PfCSP mAb combinations to their respective PfCSP mAb + VRC01 controls using the Kruskal-Wallis test (dotted lines). **(B)** MFI of Pb-PfCSP-SPZ bound by 2 μg/mL L9-AF750 when co-incubated with varying concentrations (0.032–100 μg/mL) of unlabeled CIS43, L9, or 317. **(C)** Liver burden in mice 40 hours post-challenge (n = 5/group) mediated by L9, CIS42, F10, and mAb4 alone (50 μg) and in combination (25+25 μg) administered 2 hours before IV challenge with 2,000 Pb-PfCSP-SPZ. *P*-values were determined by comparing mAb combinations to L9 alone using the two-tailed Mann-Whitney test. **(D)** MFI of Pb-PfCSP-SPZ bound by 2 μg/mL L9-AF750 when co-incubated with varying concentrations (0.032–100 μg/mL) of unlabeled CIS42, F10, or mAb4. **(E)** MFI of Pb-PfCSP-SPZ (filled squares) and PfSPZ (open squares) bound by 2 μg/mL L9-A750 when co-incubated with 100 μg/mL of specified unlabeled mAb. *P*-values were determined by comparing PfCSP mAbs to VRC01 using the Kruskal-Wallis test. (A, C): lines represent geometric mean. (B, D): dotted lines represent L9-AF750 co-incubated with 100 μg/mL unlabeled VRC01; data are representative of two independent experiments. (A, C, E): ***, p<0.001; **, p<0.01; *, p<0.05; ns (not significant), p>0.05.

neutralizing NPDP-, NVDP-, and NANP-preferring mAbs (CIS42, F10, and mAb4 respectively; Fig 1 and S1 Table) would result in improved protection compared to L9 alone. L9 alone was significantly more protective than L9 + CIS42 and L9 + F10 and provided protection comparable to L9 + mAb4 (Fig 6C). As in Fig 6B, CIS42, F10, and mAb4 all lowered L9-A750 SPZ binding, with CIS42 mediating the greatest reduction (Fig 6D). As with mAb10-A750 (Fig 5C), L9-A750 SPZ binding was not significantly affected by 5D5 or all C-CSP mAbs tested (Fig 6E). Collectively, these data confirm that combining mAbs targeting different repeat epitopes does not offer improved protection *in vivo* and that mAbs directed against the repeat region, but not N- or C-CSP, antagonize the binding of other repeat mAbs.

## Combining R21 vaccination and passive repeat mAb administration increases protection

Since the SPZ binding of repeat mAbs is antagonized by other repeat mAbs (Fig 6B–6D) but not N- or C-CSP mAbs (Figs 5C and 6E and S4B), we next determined how a polyclonal PfCSP antibody response would affect the SPZ binding of repeat mAbs. The Pb-PfCSP-SPZ binding of A750-labeled CIS43, L9, or 317 were assessed in the presence of serum pooled from ten naïve mice immunized three times with R21 (virus-like particles composed of 19 NANP repeats and C-CSP fused to a single hepatitis B surface antigen [39]), fifteen naïve US adults immunized three times with irradiated PfSPZ [51], and ten children or adults with the highest FL-rCSP titers from a cohort of 758 Malian volunteers naturally exposed to malaria [52]. Serum from a malaria-naïve US adult was included as a negative control.

While malaria-naïve serum had no effect on mAb binding compared to buffer alone, serum from vaccinated or malaria-exposed individuals significantly decreased mAb binding to SPZ. Serum from R21-vaccinated mice exerted the greatest antagonistic effect, with CIS43 showing the greatest binding reduction (Fig 7A). The antagonistic effect of R21 vaccine serum was likely due to its higher overall FL-rCSP titers compared to serum from the PfSPZ Vaccine or Natural Exposure groups (Fig 7B). These data show that polyclonal PfCSP antibody responses compete with repeat mAbs for native PfCSP binding sites on SPZ *in vitro*.

To extend the analysis, we used a combined active-and-passive immunization approach to assess *in vivo* protection in mice vaccinated with R21 that also received CIS43, L9, or 317 prior to IV challenge (Fig 7C). Importantly, mice received a dose of R21 that lowered liver burden to the approximate midpoint of the assay's dynamic range. R21-vaccinated mice that received 150 µg/mouse of mAb had significantly lower liver burdens than mice that received R21 alone, but not mAb alone (Fig 7D). These data suggest that R21-induced polyclonal antibodies did not significantly perturb mAb-mediated protection *in vivo* despite competing for SPZ binding *in vitro* (Fig 7A).

Notably, when the mAb dose was lowered to 50 µg/mouse (Fig 7E) R21-vaccinated mice that received 50 µg of CIS43 or L9 had significantly lower liver burdens than naïve mice that only received 50 µg of each mAb while mice that received R21 + 317 trended towards lower liver burden than mice that received 317 only. When compared to mice that received R21 only, mice that received R21 + L9 and R21 + 317 had significantly lower liver burden while mice that received R21 + CIS43 trended towards lower liver burden. Furthermore, lowering the mAb dose to 25 µg/mouse (Fig 7F) largely confirmed the 50 µg/mouse data, with the R21 + mAb combinations providing greater protection than mAb alone and trending towards greater protection than R21 alone. Together, these data show that potent repeat mAbs act cooperatively with vaccine-induced polyclonal PfCSP antibody responses against the repeat region and C-CSP to provide improved protection against SPZ challenge in mice despite competing for native PfCSP epitopes on SPZ.

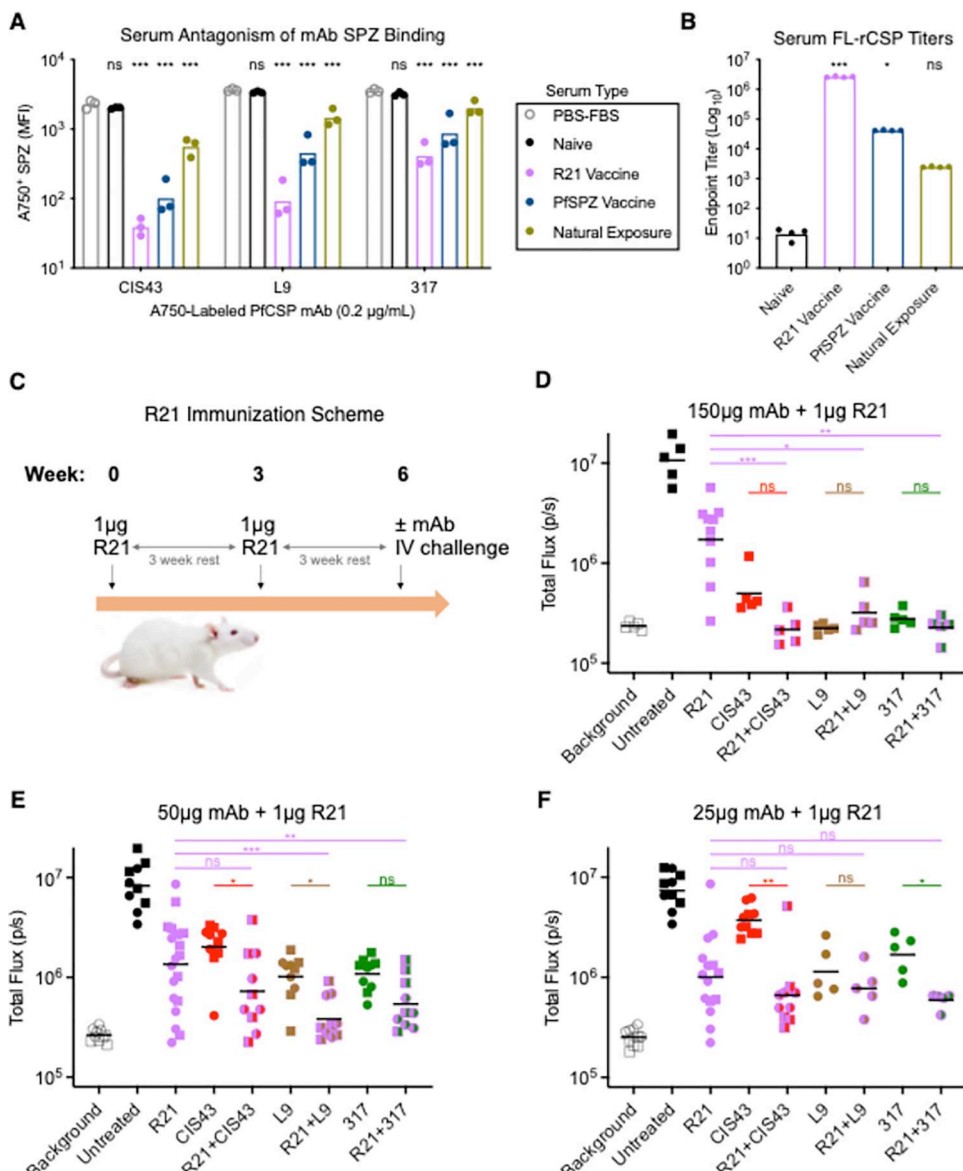

**Fig 7. Combining R21 vaccination and passive transfer of PfCSP repeat mAbs provides improved protection against malaria. (A)** MFI of Pb-PfCSP-SPZ bound by 0.2 μg/mL CIS43-, L9-, or 317-AF750 when co-incubated with PBS + 10% FBS (PBS-FBS) or 1:5 diluted serum pooled from either one naïve US adult volunteer, ten naïve mice immunized 3x with 1 μg R21 + adjuvant, fifteen naïve US adults immunized 3x with 9x10⁵ irradiated PfSPZ, and ten Malian children/adults naturally exposed to malaria. For each PfCSP repeat mAb, *P*-values were determined by comparing each serum type to the PBS-FBS control using a two-way ANOVA with Bonferroni's post-hoc correction. **(B)** Endpoint titers of pooled serum from A binding to FL-rCSP measured by ELISA. *P*-values were determined by comparing serum types to naïve serum using the Kruskal-Wallis test. **(C)** R21 immunization scheme in normal mice, which received two intramuscular injections of 1 μg R21 + adjuvant at three week intervals prior to passive transfer of PfCSP repeat mAbs two hours before IV challenge with 2,000 Pb-PfCSP-SPZ. **(D-F)** Liver burden 40 hours after IV challenge with 2,000 Pb-PfCSP-SPZ in mice (n = 5-20/group; 50μg/25μg data in E/F were combined from two experiments, circles and squares) immunized with 1 μg R21 alone; mice administered CIS43, L9, and 317 alone (D, 150 μg; E, 50 μg; F, 25 μg); and mice immunized with 1 μg R21 and administered CIS43, L9, and 317 (D, 150 μg; E, 50 μg; F, 25 μg). *P*-values were determined by comparing each R21 + mAb combination to the R21 alone or respective mAb alone groups using the Kruskal-Wallis test. (A-B, D-F): ***, p<0.001; **, p<0.01; *, p<0.05; ns (not significant), p>0.05.

Given the improved protection observed with combining R21 and repeat mAbs, we also wanted to determine the protective efficacy of combining R21 with C-CSP mAbs *in vivo*. mAb15 and R2 bound to distinct epitopes on FL-rCSP (Fig 2B), while various repeat mAbs were confirmed to potentiate the SPZ binding of mAb15 and L15 (Figs 4 and S3). Thus, R21 was combined with this subset of three C-CSP mAbs. Liver burdens in mice that received R21 and 300 μg/mouse of C-CSP mAbs were not significantly different than mice that received R21 alone (S6 Fig). These data show that, even when combined with polyclonal antibody responses against the repeat region and C-CSP, relatively high doses (300 μg/mouse) of C-CSP mAbs do not neutralize SPZ *in vivo*.

## Discussion

Here, using a large panel of thirteen human C-CSP-specific mAbs, we show that C-CSP mAbs bind FL-rCSP but do not bind or neutralize SG Pb-PfCSP-SPZ. These data are consistent with prior studies which reported that two human C-CSP-specific mAbs neither bound nor neutralized SG Pb-PfCSP-SPZ *in vitro* and *in vivo* [32] and that native PfCSP on SPZ is conformationally labile [22,33], which likely results in the masking of C-CSP on SG-SPZ from antibody recognition. It should be noted that the FL-rCSP and C-CSP constructs used in this study were expressed in mammalian cells and thus may be differentially glycosylated compared to native PfCSP on SPZ; however, a previous study found no impact of mammalian-expressed FL-rCSP or C-CSP on binding of 1710 [32].

Additionally, we show that several repeat PfCSP mAbs with different target epitopes and *in vivo* protective potencies comparably potentiate the SG-SPZ binding of C-CSP mAbs *in vitro*. These data clarify a previous report that PfSPZ immunization in humans preferentially induces antibody responses against the immunodominant repeat region and that subdominant C-CSP-specific responses are only expanded after subsequent boosts [53]. Specifically, repeat antibodies induced after the prime likely induce conformational changes in PfCSP that improve the accessibility of C-CSP to immune recognition, resulting in the expansion of C-CSP-specific responses upon subsequent PfSPZ immunizations. Notably, only a subset of repeat mAbs (i.e., 311 and CIS43) significantly changed the conformation of FL-rCSP, underscoring the discrepancies between recombinant versus native PfCSP and the need for new techniques to interrogate the native structure of PfCSP on SPZ. Collectively, these data suggest that C-CSP antibodies require the presence of repeat antibodies to bind native PfCSP on SPZ.

The potentiation of C-CSP mAb SPZ binding by repeat mAbs *in vitro* did not translate into improved protection *in vivo*. This lack of cooperative neutralization could be due to all C-CSP mAbs in the panel recognizing non-neutralizing epitopes or to potential limitations in this study's assays and/or models. Specifically, the human IgG$_1$ vector used to express mAbs in this study may be inefficient at activating Fc receptors or fixing complement in normal B6 mice and the transgenic Pb-PfCSP-SPZ used for challenges in this study may not perfectly model wild-type PfSPZ. Several studies have reported that C-CSP antibodies induced by RTS,S immunization are correlated with vaccine efficacy in phase 3 trials [15,26,27] and such antibodies can mediate human complement fixation and phagocytosis by human monocyte cell lines *in vitro* [24,25]. Thus, C-CSP + repeat mAb combinations might prove more protective compared to each mAb class alone if tested via *in vitro* studies with human cell lines, *in vivo* studies in mice with humanized Fc receptors, or in controlled human malaria infection studies.

It remains possible that polyclonal antibodies against the repeat region and C-CSP induced by vaccination may provide improved *in vivo* protection. The heterogeneous antibody response induced by RTS,S immunization might enable polyclonal C-CSP antibodies to

functionally inhibit SPZ via a mechanism not observed in the passive transfer of limited numbers of monoclonal antibodies shown here. While this study tested the largest panel of human C-CSP mAbs yet reported (i.e., thirteen C-CSP mAbs), this number is much lower than the hundreds of PfCSP repeat mAbs assessed for neutralization to date [17].

The other major focus of this study was determining whether combining highly potent human mAbs (CIS43, L9, and 317) that target the three different tetrapeptides in the repeat region would provide improved protection compared to each mAb alone. This is an important issue for the clinical development of PfCSP mAbs for malaria prophylaxis, as targeting two distinct PfCSP epitopes could lead to improved protection against malaria as has been seen with viral infections like HIV-1, Ebola, and SARS-CoV-2 [37,54,55]. A recent Phase 1 clinical trial showed that administration of CIS43 was safe and protected all nine CIS43-treated volunteers following controlled human malaria infection [56]. A similar clinical trial for L9 has recently been initiated; thus, it is possible that CIS43, L9, and other protective PfCSP mAbs might be combined in the future.

Here, combining CIS43, L9, and 317 (i.e., three of the most potent NPDP-, NVDP-, and NANP-preferring human mAbs described to date) did not result in improved protection compared to either mAb alone. Similarly, combining L9 with three poorly neutralizing NPDP-, NVDP-, and NANP-preferring human mAbs (CIS42, F10, and mAb4 respectively) conferred less protection than L9 alone. While it is possible that other combinations of repeat mAbs not tested in this study may provide superior protection compared to a single repeat mAb, our data demonstrating that the SPZ binding of a potent repeat mAb (i.e., L9) is antagonized by other repeat mAbs *in vitro* indicate that the efficacy of repeat mAb combinations may be limited by competition between mAbs that promiscuously cross-react with adjacent epitopes which share common residues [40,49]. Conversely, the observation that N- and C-CSP mAbs had no effect on repeat mAb SPZ binding even at a 50-fold excess suggests that mAbs targeting non-repeat domains may behave non-competitively when combined with repeat mAbs.

Interestingly, the SPZ binding of the NVDP-preferring mAb L9 was more potently antagonized by NPDP-preferring mAbs (CIS43, CIS42) and NANP-preferring mAbs (317, mAb4) than by NVDP-preferring mAbs (L9, F10). This data is somewhat counterintuitive, as one might expect L9 to most potently antagonize itself. A possible explanation is that the binding of CIS43 and CIS42 to NPDP may induce conformational changes in the repeat region [21] that lower the binding of L9 to NVDP/NANP repeats. 317 and mAb4 have higher affinity for NANP repeats than L9 (S1 Table) and there are more NANP than NVDP repeats in PfCSP (38 vs. 4, respectively); thus, 317 and mAb4 are likely outcompeting L9 for more epitopes in the 38 NANP repeats and thus have a greater effect on the SPZ binding of L9 than L9 and F10, which compete for fewer epitopes in the 4 NVDP repeats. These data highlight the limitations of extrapolating mAb recognition of native PfCSP epitopes based on peptides or recombinant protein and the importance of developing techniques to study mAb binding to SPZ at the atomic scale.

Remarkably, despite the lack of improved protection from combining repeat + C-CSP or repeat + repeat mAbs, combining R21 vaccination with potent repeat mAbs provided improved *in vivo* protection compared to R21 or each mAb alone. However, combining R21 with non-neutralizing C-CSP mAbs did not provide any additional protection. These data highlight the limitations of predicting *in vivo* protection against SPZ challenge based on *in vitro* binding assays. RTS,S- and R21-mediated protection is thought to be mediated primarily by antibodies, though both have been shown to induce CD4+ T cell responses [39,57]. Overall, these data suggest that vaccine-induced polyclonal repeat- and C-CSP-specific antibodies may interact cooperatively with (or at least do not lower the efficacy of) potent repeat mAbs to neutralize SPZ *in vivo* and that passive administration of protective PfCSP mAbs in RTS,S- or R21-vaccinated individuals will provide enhanced protection against malaria.

Overall, this study has implications for the optimal implementation of PfCSP mAbs to prevent malaria. No protective advantage was observed from combining the neutralizing human PfCSP repeat mAbs used in this study, which are the most potent reported to date [40], indicating that each mAb should be used as stand-alone interventions. However, our data showing that potent repeat mAbs can enhance protection in mice given sub-optimal doses of R21 suggest that passive transfer of PfCSP mAbs could be used to boost protection in vaccinated infants.

## Materials and methods

### Ethics statement

All mouse research was performed according to National Institutes of Health guidelines for use and care of live animals approved by the institutional animal care and use ethics committees of the Vaccine Research Center (Animal Study Protocol VRC-17-702).

### Human clinical specimens

Clinical specimens (i.e., peripheral blood mononuclear cells and serum) were derived from malaria-naïve, healthy adults in the VRC 314 clinical trial (https://clinicaltrials.gov/; NCT02015091), a multi-institution, phase 1, open-label, dose-escalation trial with controlled human malaria infection designed to assess the safety, immunogenicity, and protective efficacy of the Sanaria PfSPZ Vaccine administered by intravenous or intramuscular injection [51]. Serum was also collected from a cohort study conducted in the rural village of Kalifabougou, Mali [52,58].

### Production of recombinant monoclonal antibodies

C-CSP-specific mAbs were isolated in a previous study using a fluorescently-labeled FL-rCSP probe to sort probe-specific memory B cells from cryopreserved peripheral blood mononuclear cells and immunoglobulin variable region sequences were cloned into human $IgG_1$ and $Ig_\kappa/Ig_\lambda$ plasmids as previously described [40]. Sequence analysis was performed using The International Immunogenetics Information System (IMGT, http://www.imgt.org/) and phylogenetic trees were generated using Geneious Prime. The sequences of 1710 [32] were retrieved from the Protein Data Bank and cloned into the aforementioned plasmids. F10 was cloned from a plasmablast derived from the same subject that yielded mAb L9 and is a clonal relative of L9. All sequences of repeat-specific mAbs except F10 were previously described [20,21,40,41,44,59] and all were cloned into the aforementioned plasmids. Matched heavy and light chain plasmids were co-transfected into Expi293 cells using the ExpiFectamine 293 Transfection Kit (Thermo Fisher Scientific) and incubated at 37°C, 8% $CO_2$ for 6 days. Supernatants were harvested and purified using rProtein A Sepharose Fast Flow resin (GE Healthcare) and buffer exchanged with 1X PBS (pH 7.4) before being concentrated using Amicon Centrifugal Filters (Millipore). Purified mAb concentrations were determined using a Nanodrop spectrophotometer (Thermo Fisher Scientific).

### Production of recombinant PfCSP proteins and peptides

Recombinant protein corresponding to full-length PfCSP from the 3D7 clone of the NF54 isolate (PfCSP_3D7; PlasmoDB ID: PF3D7_0304600) was generated as previously described [40]. N-CSP, $(NANP)_9$, and overlapping 15mer peptides 21, 22, 29, and 67–88 were created by Genscript, as previously described [21,40]. For the recombinant C-CSP construct, a codon-optimized sequence encoding PfCSP_3D7 amino acids 295–374 was cloned in-frame with a

sequence encoding tissue plasminogen activator leader peptide [60] on the 5' end and a Gly-SerGlySerGly linker followed by a His8 tag on the 3' end. The construct was inserted into the pTT3 vector [61] for expression in FreeStyle 293F cells (Thermo Fisher), which were cultured and processed as previously described [62]. Briefly, cells were transfected using PEI MAX (Polysciences, Inc) and subsequently cultured for 5 days. Upon harvesting by centrifugation, the supernatant was treated by addition of $NaN_3$ (0.02%, final concentration) and NaCl (+350 mM), and protein was purified by immobilized metal affinity chromatography followed by size-exclusion chromatography over a calibrated HiLoad 16/600 Superdex 200 pg column (GE Healthcare). Peak fractions were pooled, flash-frozen in liquid nitrogen, and stored at -20°C until use.

## ELISA

Immulon 4HBX flat bottom microtiter plates (Thermo Fisher Scientific) were coated with 100 μl per well of antigen (0.5 μg/mL for FL-rCSP, N-CSP, (NANP)$_9$, and C-CSP; 0.1 μg/mL for 15mer peptides) in bicarbonate buffer overnight at 4°C. Coated plates were blocked with 200 μl of PBS + 10% FBS for 2 hrs at room temperature, followed by incubation with 100 μl of anti-PfCSP or control mAbs at varying concentrations ($5x10^{-7}$–5.0 μg/mL, 10-fold serial dilutions) or serum (1:20 dilution followed by 10-fold serial dilutions) for 2 hrs at 37°C and incubation with 100 μl/well of 0.1 μg/mL HRP-conjugated goat anti–human IgG (Bethyl Laboratories) or anti-mouse IgG (Santa Cruz Biotech) for 1 hr at room temperature. Plates were washed six times with PBS-Tween between each step. Samples were then incubated for 10 min with 100 μL 1-Step Ultra TMB-ELISA Substrate (Thermo Fisher Scientific) prior to the addition of stopping solution (2N sulfuric acid, 100 μl/well). The optical density at 450 nm ($OD_{450}$) was measured for each plate. To generate AUC heat map, $OD_{450}$ values were plotted against log-transformed mAb concentrations in GraphPad Prism (version 7.0) and the AUC analysis function was used to calculate the Peak Area with a baseline of Y ($OD_{450}$) = 0.2. The average AUC values for binding to each antigen from 2–4 replicate experiments were plotted as a Heat Map in Prism. Endpoint titers for the serum ELISAs were determined by nonlinear regression and interpolation of the log-transformed serum dilutions (GraphPad Prism, version 7.0).

## Epitope binning

Epitope binning experiments were performed with the Carterra LSA. A HC30M chip (Carterra) was primed with filtered and degassed 25mM 2-(N-morpholino)ethanesulfonic acid (MES) buffer supplemented with 0.05% Tween-20 (Thermo Fisher Scientific). The chip was activated with a mixture of 400 mM 1-ethyl-3- (3-dimethylaminopropyl)carbodiimide hydrochloride (EDC) and 100 mM N-hydroxysuccinimide (NHS) (Thermo Fisher Scientific). Next, 10 μg/mL of each mAb in pH 4.5 acetate buffer was directly coupled to discrete spots on the chip, followed by blocking with 1M ethanolamine (pH 8.5). The chip was then primed with HEPES-buffered saline Tris-EDTA (HBSTE) + 0.05% BSA. Monomeric 200 nM FL-rCSP was added to the antibody spots, followed by addition of 10 μg/mL of the sandwiching mAb. Regeneration after incubation with each sandwiching antibody was performed with 10 mM glycine pH 2.0. Binning data were analyzed using the Epitope Software (Carterra).

## Isothermal titration calorimetry

All isothermal titration calorimetry studies were performed using a VP-ITC microcalorimeter (Malvern Panalytical), with FL-rCSP and mAbs in PBS (pH 7.4). Each mAb solution (~40 μM; expressed per antigen binding site) was injected in 5 or 7 μl aliquots into the calorimetric cell

containing FL-rCSP (~0.4 μM) at the specified temperatures (15–35°C). The exact concentrations of the reactants in each experiment were determined by their absorbance at 280 nm. The heat evolved upon each mAb injection was determined from the integral of the calorimetric signal. The heat associated with binding to FL-rCSP was obtained by subtracting the heat of dilution from the heat of reaction. The individual heats were plotted against the molar ratio, and the enthalpy change, $\Delta H$, the association constant, $K_a$ (the dissociation constant, $K_d = 1/K_a$) and the stoichiometry (valency of antigen binding sites), $N$, were obtained by nonlinear regression of the data to a model that takes into account the binding to either one or two sets of sites with different binding affinities [63]. Gibbs energy, $\Delta G$, was calculated from the relation $\Delta G = -RT\ln K_a$, where $R$ is the universal gas constant, (1.987 cal/(K × mol)) and $T$ the absolute temperature in Kelvin. The entropy contribution to Gibbs energy, $-T\Delta S$, was calculated from the known relation $\Delta G = \Delta H—T\Delta S$. The results were expressed per mole of antigen binding sites and the stoichiometry, $N$, denotes the number of antigen binding sites per mole of FL-rCSP.

## Fluorescent antibody labeling

Antibodies were fluorescently labeled with the SAIVI Rapid Antibody Labeling Kit, Alexa Fluor 750 (A750), according to the manufacturers' instructions (Thermo Fisher Scientific). In brief, purified mAbs were mixed with reconstituted kit components, incubated for 1 hr at room temperature, and purified over the SAIVI column. The absorbance correction factor (CF = 0.034) and extinction coefficient ($\epsilon$ = 270,000 $M^{-1}cm^{-1}$ at 752 nm) were used to determine mAb concentration and A750 degree of labeling (DOL) ratios (~2–4) using a Nanodrop spectrophotometer. To ensure binding was not dramatically changed by A750 labeling, the binding of all A750-labeled mAbs was compared to original unlabeled mAbs by FL-rCSP ELISA.

## Sporozoites

Transgenic *P. berghei* (strain ANKA 676m1c11, MRA-868) expressing full-length *P. falciparum* CSP and a green fluorescent protein/luciferase fusion protein (Pb-PfCSP-GFP/Luc-SPZ) were obtained as previously described [64]. *Anopheles stephensi* Nijmegen strain mosquitoes were infected with *P. berghei* malaria-infected Balb/c mice or with *P. falciparum* NF54 gametocytes using membrane feeding. Pb-PfCSP-GFP/Luc-SPZ were dissected from salivary glands 20–24 days post-infection; *P. falciparum* sporozoites were dissected from salivary glands 16–18 days post-infection. Sporozoites were isolated into either ice-cold PBS with or without 10μM E-64 protease inhibitor (Sigma-Aldrich) to prevent PfCSP proteolytic cleavage or Leibovitz's L-15 medium (Sigma-Aldrich). The infected salivary glands were homogenized gently by passing them 15 times through a syringe with a 28G needle or grinder with pestles for tissue. The salivary gland homogenate was passed through a 40 μm cell strainer (Pluriselect, USA) by gravity. The sporozoites were counted in a hemocytometer and placed on ice.

## Flow cytometric measurement of mAb binding to sporozoites

Freshly isolated Pb-PfCSP-GFP/Luc-SPZ or PfSPZ in PBS ± E-64 were stained with SYBR Green (10,000X concentrate; Thermo Fisher Scientific) diluted 1:2,000 in PBS for 30 min at 4°C, washed twice, and ~8,000 SPZ were aliquoted to each well of a 96-well V-bottom plate (50 μl/well). For measurement of single mAb binding, SPZ were incubated for 30 min at 28°C with anti-PfCSP or control mAbs in PBS + 10% FBS (PBS-FBS), washed twice with 200 μl PBS-FBS, stained for 20 minutes at 4°C with goat anti-human IgG-Alexa Fluor 647 secondary antibody (Thermo Fisher Scientific) diluted 1:1,000 in PBS-FBS, washed once with 200 μl

PBS-FBS, and fixed in 200 μL PBS with 0.5% paraformaldehyde (PBS-PFA). For measurement of co-incubated A750-labeled and unlabeled mAbs, SPZ were incubated with specified concentrations of both mAbs for 30 min at 28˚C, washed twice with 200 μl PBS-FBS, and fixed in 200 μL PBS-PFA. For measurement of co-incubated A750-labeled mAbs and serum, SPZ were incubated with 0.2 μg/mL of A750-labeled mAbs and serum diluted 1:5 in PBS-FBS for 30 min at 28˚C, washed twice with 200 μl PBS-FBS, and fixed in 200 μL PBS-PFA. Following fixation, events were acquired on a modified LSR II (BD Biosciences). FlowJo v.10 was used to determine the median fluorescence intensity in A750 of SYBR Green-positive events corresponding to SPZ.

### R21 immunizations in mice

R21 immunogen [39] was diluted in sterile PBS to 1 μg with 33.3 μL of army liposomal formula Q (ALFQ; liposomal adjuvant formulation containing monophosphoryl lipid A and QS-21 [65]) in a final volume of 50 μL. Female 6- to 8-week-old B6(Cg)-Tyrc-2J/J albino mice were immunized intramuscularly in the quadriceps at 3 week intervals with 50 μL of R21 + ALFQ. Passive transfer of mAbs prior to Pb-PfCSP-SPZ challenge were performed 3 weeks following the last immunization (i.e., week 6), as described below.

### *In vivo* mouse challenge studies with Pb-PfCSP-GFP/Luc-SPZ

Specified amounts of anti-PfCSP or control mAbs diluted in sterile filtered PBS (200 μl/mouse) were injected into the tail veins of female 6- to 8-week-old B6(Cg)-Tyrc-2J/J albino mice (The Jackson Laboratory). For intravenous challenge, 2,000 freshly harvested Pb-PfCSP-GFP/Luc-SPZ in Leibovitz's L-15 medium (Sigma-Aldrich) were injected into the tail vein 2 hours after mAb administration. For intradermal challenge, 5,000 freshly harvested Pb-PfCSP-GFP/Luc-SPZ in L-15 medium were injected into the paw 24 hours after mAb administration. Liver burden (40–42 hours post-challenge) or parasitemia (day 5 post-challenge) were respectively quantified by intraperitoneally injecting mice with 150 μl of D-Luciferin (30 mg/mL), anesthetizing them with isoflurane, and measuring the bioluminescent radiance (total flux; photons/sec) expressed by Pb-PfCSP-GFP/Luc-SPZ using the IVIS Spectrum *in vivo* imaging system (PerkinElmer) 10 minutes after luciferin injection. Total flux in regions of interest were quantified using Living Image 4.5 software (PerkinElmer).

### Statistical analysis

Unless otherwise indicated, all data were plotted using GraphPad Prism, version 7.0. Statistical tests used, exact value of n, and what n represents can be found in figure legends. For the ITC stoichiometry data, errors with 95% confidence were estimated from the fits of the data.

### Supporting information

**S1 Table. Classification of monoclonal antibodies (mAbs) used in this study.** mAb designation, original isotype (IgG vs. IgM; all mAbs used in this study were expressed as IgG$_1$), species (human vs. mouse), target antigen, BLI apparent avidity for select PfCSP 15mer repeat peptides (peptide 21 NPDPNANPNVDPNAN, peptide 22 NANPNVDPNANPNVD, peptide 29 NANPNANPNANPNAN), SPZ binding (none, low, high), SPZ neutralization (none, very low, low, moderate, high), and references used to classify mAbs. Undet., undetectable; n/a, not applicable.
(DOCX)

**S2 Table. Gene usage and variable region sequences of 13 human C-CSP mAbs.** V-gene families, somatic hypermutation (% SHM), and amino acid sequences of the heavy chains ($V_H$) and light chains ($V_L$) of the 13 C-CSP mAbs used in this study. L4-L40 and R1-R7 were cloned from the same naïve volunteer immunized with whole PfSPZ; mAb15 and 1710 were cloned from separate volunteers also immunized with whole PfSPZ. The % SHM of 1710 was not included as the original nucleotide sequence could not be retrieved.
(DOCX)

**S3 Table. Normalized heat map of mAb binding competition to FL-rCSP by SPR.** Vertically-listed mAbs are ligands; horizontally-listed mAbs are analytes. Green boxes indicate positive values (no competition); red boxes indicate negative values (competition); blacked out boxes indicate self-self competition and were thus excluded. Several mAbs (L7, L15, L20, R1, mAb10) were excluded due to poor ligand and/or analyte binding to FL-rCSP.
(DOCX)

**S1 Fig. Phylogenetic trees and peptide mapping of C-CSP mAbs. (A)** Phylogenetic trees of the heavy chain (left) and light chain (right) sequences of the C-CSP mAbs in this study. 1710 was not included as its original nucleotide sequence could not be retrieved. **(B)** Binding of varying concentrations of thirteen pooled C-CSP-specific mAbs (L4–1710) to 15mer overlapping peptides numbered 67–88 as determined by ELISA. Optical density at 450 nm ($OD_{450}$) is plotted; peptide sequences are depicted. C-CSP and the NANP-containing peptide 29 were included as positive and negative controls, respectively.
(TIFF)

**S2 Fig. CIS43 and 311 induce conformational changes in FL-rCSP. (A)** ITC plots of CIS43, L9, 311, and 317 IgG binding to FL-rCSP at indicated temperatures (15–35˚C). Top, dQ/dt (change in heat flow, Q, as a function of time, t). Bottom, the integrated heat associated with each IgG injection shown as a function of the molar ratio between IgG antigen binding sites and FL-rCSP in the calorimetric cell. The red line represents the result from best nonlinear least squares fit of the data. Enthalpy values of the first and second binding events ($\Delta H_1$ and $\Delta H_2$, respectively) are shown. **(B)** Plots of the enthalpy changes $\Delta H$ as a function of temperature (15–35˚C) for the binding of each mAb to the first ($\Delta H_1$, left panel) and second ($\Delta H_2$, right panel) sets of sites, respectively. The equation of each line (y = mx + b) and correlation coefficient (R) are depicted. The change in heat capacity ($\Delta Cp$) associated with each binding event is equal to m (the slope of the line) and is underlined in the equation of the line and depicted in the plots for each mAb.
(TIFF)

**S3 Fig. PfCSP repeat mAbs potentiate the SPZ binding of C-CSP mAbs. (A)** Representative flow cytometry plots depicting 20 µg/mL mAb15-A750 (top panel) or L15-A750 (bottom panel) binding to SG Pb-PfCSP-SPZ in the presence of 20 µg/mL unlabeled VRC01, 5D5, CIS43, and repeat mAb mix (mAb mix, CIS43-317 in B). Percentages of mAb15$^+$ or L15$^+$SPZ are shown. **(B)** Percentage and MFI of Pb-PfCSP-SPZ (filled squares) and PfSPZ (open squares) bound by 20 µg/mL L15-A750 when co-incubated with 20 µg/mL of specified unlabeled mAb. *P*-values were determined by comparing PfCSP mAbs to VRC01 using the Kruskal-Wallis test. **(C)** Binding of four A750-labeled C-CSP mAbs (20 µg/mL) to Pb-PfCSP-SPZ (filled symbols) or PfSPZ (open symbols) when co-incubated with unlabeled VRC01 (20 µg/mL), with or without the protease inhibitor E-64. *P*-values were determined by comparing -E-64 to +E-64 for each C-CSP mAb using a two-way ANOVA with Sidak's post-hoc correction. (B, C): ***, p<0.001; **, p<0.01; *, p<0.05; ns (not significant), p>0.05.
(TIFF)

**S4 Fig. mAb15 and L15 neither potentiate the SPZ neutralization nor antagonize the SPZ binding of CIS43 and 311. (A)** Liver burden in mice (n = 5/group; line indicates geometric mean) 40 hours post-challenge mediated by indicated mAb combinations (CIS43 and 311, 50 μg; VRC01, mAb15, L15, 300 μg) administered 2 hours before IV challenge with 2,000 Pb-PfCSP-SPZ. *P*-values were determined by comparing repeat mAb + VRC01 to untreated control (black values) or C-CSP mAb + repeat mAb to repeat mAb + VRC01 (colored values) using the Kruskal-Wallis test with Dunn's post-hoc correction. *, p<0.05; ns (not significant), p>0.05. **(B)** MFI of Pb-PfCSP-SPZ bound by 2 μg/mL CIS43-A750 when co-incubated with 100 μg/mL of specified unlabeled mAb.
(TIFF)

**S5 Fig. Combinations of CIS43, L9, and 317 do not cooperatively neutralize SPZ *in vivo*. (A-C)** Liver burden 40 hours after IV challenge with 2,000 Pb-PfCSP-SPZ in mice (n = 5-10/group; 50 μg data in E was combined from two experiments, circles and squares) that received CIS43, L9, and 317 alone (A, 100 μg; B, 50 μg; C, 25 μg) or in combination (A, 50+50 μg; B, 25+25 μg; C, 12.5+12.5 μg). Black lines represent geometric mean. *P*-values were determined by comparing mAb combinations to L9 or 317 alone using the two-tailed Mann-Whitney test. ***, p<0.001; **, p<0.01; *, p<0.05; ns (not significant), p>0.05.
(TIFF)

**S6 Fig. Combining R21 vaccination and PfCSP C-CSP mAbs does not provide improved protection.** Liver burden 40 hours after IV challenge with 2,000 Pb-PfCSP-SPZ in mice (n = 5-10/group) immunized with 1 μg R21 alone; mice administered 300 μg of C-CSP mAbs (mAb15, L15, R2) or 25 μg of CIS43 alone; and mice immunized with 1 μg R21 and administered 300 μg of C-CSP mAbs (mAb15, L15, R2) or 25 μg of CIS43. *P*-values were determined by comparing each R21+mAb combination to the R21 alone or respective mAb alone groups using the Kruskal-Wallis test. ***, p<0.001; **, p<0.01; *, p<0.05; ns (not significant), p>0.05.
(TIFF)

## Acknowledgments

We thank B. Kim Lee Sim and Stephen L. Hoffman (Sanaria, Inc.) for providing the PfSPZ Vaccine; David Ambrozak (NIH Vaccine Research Center) for assistance with sorting memory B cells; Vladimir Vigdorovich, Nicholas Dambrauskas, and Noah Sather (Seattle Children's Hospital) for providing the recombinant C-CSP protein and plasmid; Alexander Anderson and Gary Matyas (Walter Reed Army Institute of Research) for providing the ALF-Q adjuvant; Alexandra Spencer and Adrian Hill (University of Oxford) for providing the R21 vaccine; Shanping Li and Peter Crompton (NIH Laboratory of Immunogenetics) for providing serum from Malians living in malaria-endemic regions.

## Author Contributions

**Conceptualization:** Lawrence T. Wang, Azza H. Idris, Joseph R. Francica, Robert A. Seder.

**Formal analysis:** Lawrence T. Wang, Lais S. Pereira, Arne Schön, Joshua Tan.

**Funding acquisition:** Carolina Barillas-Mury, Robert A. Seder.

**Investigation:** Lawrence T. Wang, Lais S. Pereira, Patience K. Kiyuka, Arne Schön, Neville K. Kisalu, Rachel Vistein, Marlon Dillon, Brian G. Bonilla, Joshua Tan.

**Methodology:** Lawrence T. Wang, Lais S. Pereira, Joshua Tan, Joseph R. Francica.

**Resources:** Alvaro Molina-Cruz, Carolina Barillas-Mury.

**Supervision:** Azza H. Idris, Joseph R. Francica, Robert A. Seder.

**Validation:** Lawrence T. Wang, Lais S. Pereira, Arne Schön, Azza H. Idris, Joseph R. Francica, Robert A. Seder.

**Visualization:** Lawrence T. Wang, Lais S. Pereira, Arne Schön.

**Writing – original draft:** Lawrence T. Wang, Azza H. Idris, Joseph R. Francica, Robert A. Seder.

**Writing – review & editing:** Lawrence T. Wang, Lais S. Pereira, Patience K. Kiyuka, Arne Schön, Neville K. Kisalu, Rachel Vistein, Marlon Dillon, Brian G. Bonilla, Alvaro Molina-Cruz, Carolina Barillas-Mury, Joshua Tan, Azza H. Idris, Joseph R. Francica, Robert A. Seder.

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
