## [Decision Letter · Decision Letter 0]

20 Aug 2021

Dear Dr. Seder,

Thank you very much for submitting your manuscript "Binding and protection with combinations of human monoclonal antibodies against the repeat region and C-terminal domain of the Plasmodium falciparum circumsporozoite protein" for consideration at PLOS Pathogens. As with all papers reviewed by the journal, your manuscript was reviewed by members of the editorial board and by several independent reviewers. In light of the reviews (below this email), we would like to invite the resubmission of a significantly-revised version that takes into account the reviewers' comments.

When revising your manuscript I particularly draw your attention to comments from Reviewers 2 and 3. Its important that you respond to the limitations in the approaches used that were raised by the reviewers, including limitations of mice for modelling human immunity, potential mechanisms of immunity are not limited to neutralization, functions of MAbs may be different to polyclonal antibodies, and MAbs were isolated from naïve adults who received a live-attenuated vaccine (responses in malaria exposed subjects, children, or after RTS,S vaccine may be quite different). Several of these points can be addressed without the need for substantial new experimental data.

You may also consider Reviewer 3 suggestion to shift the emphasis away from the lack of protection by CT MAbs (given the limitations noted) onto other interesting and potentially important findings you present.

We cannot make any decision about publication until we have seen the revised manuscript and your response to the reviewers' comments. Your revised manuscript is also likely to be sent to reviewers for further evaluation.

Sincerely,

James G. Beeson, MBBS, PhD

Guest Editor

PLOS Pathogens

James Kazura

Section Editor

PLOS Pathogens

Kasturi Haldar

Editor-in-Chief

PLOS Pathogens

orcid.org/0000-0001-5065-158X

Michael Malim

Editor-in-Chief

PLOS Pathogens

orcid.org/0000-0002-7699-2064

When revising your manuscript I particularly draw your attention to comments from Reviewers 2 and 3. Its important that you respond to the limitations in the approaches use that were raised by the reviewers, including limitations of mice in modelling human immunity, potential mechanisms of immunity, functions of MAbs may be different to polyclonal antibodies, and MAbs were isolated from naïve adults who received a live-attenuated vaccine (responses in malaria exposed subjects, children, or after RTS,S vaccine may be quite different). Several of these points can be addressed without the need for substantial new experimental data.

You may also consider Reviewer 3 suggestion to shift the emphasis away from the lack of protection by CT MAbs (given the limitations noted) onto other interesting and potentially important findings you present.

Reviewer's Responses to Questions

**Part I - Summary**

Reviewer #1: Wang et al. report on the in vitro sporozoite recognition and in vivo protection capacities of thirteen monoclonal antibodies (mAbs)that recognize conformational epitopes in the C-terminal domain of PfCSP (C-CSP) with or without combination with mAbs against three tetrapeptides in the PfCSP repeat region. Interestingly, they show that while the combination enhances sporozoite recognition in vitro, it does not enhance protection against challenge in a rodent model of infection. Also of note, the data reported here substantiate the existence of distinct PfCSP conformations on midgut versus salivary gland sporozoites and indicate that C-CSP mAbs are able to distinguish between the two native PfCSP conformations. Additionally, the authors show that mAbs against the repeat region of PfCSP potentiate C-CSP mAb binding to sporozoites by changing the protein’s conformation, albeit without this resulting in an increase in protection in vivo. Finally, they show that mAbs against the repeat region of PfCSP do not synergize in vivo and, unlike mAbs against the N- or C-termini of PfCSP, antagonize the binding of other repeat mAbs to sporozoites. These results are novel and relevant to the field, as they may not only shed light on PfCSP antibody-mediated protection against Plasmodium infection, but also inform on the design on vaccines against malaria. The study is well performed and the conclusions are warranted by the results. Overall this is a study worthy of publication and only a suggestion is made for its improvement.

Reviewer #2: This study by the Seder group evaluates the sporozoite binding and antiparasitic capacity of a large panel of human mAbs against different regions of P. falciparum CSP (central repeat and C-terminus) in a mouse model, and report no evidence for C-terminus antibodies, with weak binding alone that is potentiated when combined with anti-repeat mAbs. C-terminus mAbs had no neutralizing effect against sporozoite challenge in passive transfer murine experiments, alone or in combination, in contrast with previously well characterized mAbs against the repeat epitopes. Furthermore, no evidence is found for additive or synergistic neutralizing effect of combination of mAbs in contrast with other diseases, in fact they found that some anti-repeat mAbs (but not anti-C-terminus) could antagonize the effect of other anti-repeat mAbs. The considerations about binding of anti-C-terminus mAbs depending on the open vs closed conformation of the CSP, and the two-step model, are interesting and provide insight into availability (unmasking) of epitopes for proper function. The strengh and novelty relies on the detailed analysis of a panel of mAbs from humans immunized with sporozoite vaccines. The main limitation is the strong focus on the neutralization function that is not a correlate of protection in humans and the limitations of the transgenic mouse model used that may not reflect the in vivo biological processes in African individuals upon exposure and vaccination. As a consequence, a role for CSP C-terminus Abs and an enhanced protection in combination with anti-repeat Abs via other mechanisms still cannot be discarded. This could be reflected better in the title (that could include the term neutralization), abstract and discussion/conclusions.

Reviewer #3: Yang et al present a dissection of the in vitro and in vivo activities of a panel of monoclonal antibodies to the C-terminus of PfCSP, and also investigate potential for beneficial interactions between combinations of CSP-repeat-specific mAbs in in vivo protection. The study attempts to shed light upon an uncertainty about the role of C-CSP-specific antibody in protection induced by the RTS,S malaria vaccine.

The key positive finding is that antibodies to CSP repeat result in increased accessibility of the CSP C-terminus for binding of these mAbs. The remaining results are essentially negative: in particular, the authors do not find evidence of a protective effect of the C-CSP-specific antibodies, with or without the addition of CSP-repeat-specific antibodies.

Negative results can of course be very important, but in this case it is probably not possible to say whether the absence of a protective effect is due to a characteristic of these specific mAbs / these specific in vivo & in vitro assays, or whether C-CSP-specific antibodies in general lack the potential for protection. In some ways mAbs may not be the ideal tool to resolve the uncertainty about the role of C-CSP-specific antibody: it is difficult to know whether a mAb panel is representative of the range of activities which may be achieved by polyclonal antibody. Use of antigen-affinity-purified C-CSP-specific polyclonal antibody as a complementary reagent in some of the work might have been helpful. The authors neither replicate nor attempt to refute the conflicting previous observation that C-CSP-specific polyclonal Ab can inhibit sporozoite invasion, and the authors do not investigate the ability of the novel mAbs to trigger other mechanisms like phagocytosis and complement fixation, which the introduction highlights have been suggested to be involved in protection.

I think ability to draw generalizable conclusions from this work about mechanisms of vaccine-induced production will therefore be limited, unless the authors are able to more fully compare the effects of these mAbs with those of vaccine-induced C-CSP-specific pAb using a more comprehensive set of assays. I think the main contribution of this article might instead be to make available a set of new reagents to the community (although as far as I can see it is not clear whether/how the mAbs will be available). Further work is needed to clarify how these new antibodies are differentiated from each other and from previously reported mAbs (major point 1 below), and perhaps to move the emphasis of the manuscript away from the negative results. Although the conclusions are worded carefully and appropriately, the manuscript reads a little like an unsuccessful effort to find a protective effect, rather than a considered effort either to disprove the possibility of a protective effect or to fully characterise a new set of reagents.

**Part II – Major Issues: Key Experiments Required for Acceptance**

Reviewer #1: None

Reviewer #2: The major issue is the framework in which the study is presented, ie. the adequacy of the model and assumptions for generalizability of results. The in vivo protection against sporozoite challenge in mice and redout chosen may not reflect the mechanisms of in vivo protection in humans mediated by those antibodies.

- Neutralizing Abs are not a correlate of protection therefore it is possible that other mechanisms are more relevant for the tested mAbs

- Need to refer to other articles having evidence in the direction that combination of CSP repeat + C-terminus Abs having a more potent role than individually with different readouts

Reviewer #3: 1. The authors should present data to make clear whether the various C-CSP-specific mAbs are truly distinct from each other, or from the previously-published C-CSP-specific mAbs, in terms of clonal relatedness and binding competition. Sequence analysis and a binding competition assay should be relatively straightforward. Ideally, as one of the arguments for publication of this article in PLoS Pathogens rather than a more specialist journal is that it reports a panel of new reagents which may be of use to the community, the amino acid sequences of the new mAbs should be reported so that they can be produced and used by others.

2. (Modification of analysis rather than additional experiment) There is a lack of clarity about what synergism, additivity and sub-additivity really mean with respect to the action of combinations of monoclonals, and hence a lack of clarity about how to test for them (or the extent to which the unavoidable use of a complex in vivo assay restricts the feasibility of testing for them). Stemming from this, I think the conclusions / figure titles / data analysis regarding the in vivo effects of mAb combinations (Figures 3, 4, S5) should be revised.

At present there are confusing webs of stars & lines on Figures 4 and S5 (and to a lesser extent Figure 3), indicating multiple statistical comparisons on the figures looking at the in vivo effects of pairs of antibodies. In each case more clarity is needed on what the key hypothesis was, how it was tested, and how this should be interpreted.

Comparison to the untreated control isn’t really very relevant – with the exception of the non-neutralising anti-repeat mAbs in Figure 4C, it’s clear that all tested mAb treatments are protective. Instead I think the authors are trying to determine, by looking for differences between certain pairs of treatments, what can be said about whether any such differences (or the lack of them) are indicative of sub-additive, additive (i.e. independent) or synergistic interactions between the treatments.

The basic design of the in vivo experiments (comparing ‘X ug of mAb A + X ug of mAb B’ vs ’2X of A’ vs ‘2X of B’) is not really sufficient to detect anything other than a major synergistic effect (for which the effect of A+B would be better than seen with either 2A or 2B). Such synergy is not seen.

Without groups receiving X ug of A alone (not 2X), and X ug of B alone (or ideally dose response curves for X and Y alone), and some further maths to identify the expected result if A & B were acting independently, I don’t think it is possible to distinguish between weak synergy, additivity, or sub-additive (interfering) activities. I think all the data presented could be consistent with an additive effect- the combinations of half-doses of two mAbs don’t appear worse than full doses of a single mAb.

I would not advocate re-performing the experiments with additional groups, but the analysis and interpretation should be reviewed and made more precise.

**Part III – Minor Issues: Editorial and Data Presentation Modifications**

Reviewer #1: The paper is well organized, the data are clearly presented, and the results are adequately shown and discussed. However, I would suggest that the authors consider moving some of the data currently presented as Supplementary information to the main figures of the article. Specifically, I would say that the Scheme at the top of Fig. S1-A is informative enough to be shown in the main article, and that the data in Fig. S2 (or a selection thereof) and Fig. S5 are important enough to warrant their inclusion as main figures in the manuscript.

Reviewer #2: Abstract

- Indicate the source of the human mAbs

Introduction

- Acknowledge that there may be other non-neutralizing functions of sporozoite antibodies and provide references

- In addition, CSP binding (titers) correlate with neutralizing activity, however this is not a surrogate of protection and thus other functions in addition to neutralization should be considered, not only in the RTS,S context but also in naturally acquired and attenuated sporozoite immunity

- Make a more clear distinction in the literature refered as to what are human studies and what are mice studies, which vaccine, and whether tested in naive or exposed people, in adults or children.

- Review the completeness and adequacy of some key references, for example Ref 5 is a small study about booster responses to CSP but a prior larger study after primary vaccination including anti-repeat and anti-C-terminus CSP Abs is not mentioned, and is required to complement Ref 6 that only assessed anti-repeat IgG titers. Ref 18 reports association IgG2 with protection while other studies not refered report the opposite.

Results

-C-CSP-specific mAbs had limited binding to sporozoites (as said in abstract) or no binding (as said in results)?

Discussion

- There is no reference about the potential relevance of glycosilation on the C-terminus, since it was expressed in mammalian cells this may be warranted

- In general, even though the authors refer to field studies of malaria vaccines showing different results with polyclonal responses to C-terminus, it would be necessary to more clearly acknowledge the fact that the lack of consistency between their data and the field trials data could indicate limitations in their readouts and model, as mentioned above (particularly the sole focus on neutralization activity by mAbs)

Reviewer #3: 1. Given the large number of mAbs used in the study, and the non-intuitive nomenclature, it is not easy for the reader to follow what epitopes are bound by whichever mAb is under discussion at a given point. Could the authors devise some more helpful nomenclature / ‘coding’ system – e.g. use, across all figures, consistent colour-coding of mAbs by binding specificity?

2. Line 33-34 ‘did not provide enhanced protection’ requires statement of what the lack of enhancement was in comparison with (presumably C-CSP mAbs alone).

3. The author summary is not very clearly written, especially lines 49-53 – it could benefit from being re-worded.

4. Fig 1A: An AUC heat map is not a particularly common way of reporting ELISA data. Please provide further explanation of how this was produced.

5. Fig 1B: BLI data does not all appear to be well fitted by the 1:1 model (it is not explained why the red ‘model-fit’ lines are not shown for all concentrations). Ability to interpret the kinetics is compromised because, with IgG in solution, the binding will not be 1:1. Could this assay not have been run either using Fabs or with the CSP in solution, so true affinity could be measured from a true 1:1 interaction?

6. Fonts on some figures are very small indeed & it would be helpful to consider rearranging / redesigning the figures to improve legibility – eg fig 2, supp fig 1A. Similarly, some panels try to represent quite complex data sets and become quite confusing even though the key result could be shown more simply, perhaps by moving some of the information to supplementary (e.g. the gating shown in 2B could be supplementary or reduced to a single example, and the transgenic spz data sets on panels 2C & 2E are superfluous- the wt Pf data makes the point).

7. Figure S3 & associated description in results section: this biophysics data, and the conclusions drawn from it, are quite complex. I suspect many/most readers would require further explanation to grasp what is being shown. I am unable to comment on whether the data shown supports the conclusions being drawn.

8. Fig S3 axis label typo ‘enthalpty’

9. I don’t think that the data shown in Fig S4A supports the conclusion stated in the figure title. Instead it shows that L15 & mAb 15 do not potentiate spz neutralization by CIS43 or 311. The conclusion stated by the title cannot be drawn without ‘L15 only’ and ‘mAb 15 only’ conditions.

10. Lines 282-288 make sense as an investigation of one possible explanation for why a synergistic effect wasn’t seen, but don’t quite ‘fit’ at this point in the narrative - they do not support the concluding sentence of the paragraph / subsection. I think they either need to come after that sentence, or the concluding sentence needs to be modified to encompass the additional information from 282-288.

11. Fig S4B needs explanation in the legend of what the reader is meant to draw from it – it makes sense when referred to from the results section, but does not link to the title of Fig S4, or panel S4A.

12. Line 457 – make clear that the Fc was human rather than mouse IgG1 (as stated in discussion). Some further explanation of the expected ability of this Fc to recruit Fc-dependent effector functions in the in vivo studies would be helpful.

13. The logical flow of the manuscript is slightly disrupted by jump from focus on C-CSP to the investigation of combinations of anti-repeat mAbs. The finding of lack of in vivo synergy between pairs of antibodies which are competitive antagonists of each others’ binding is not very surprising. I therefore think the readability of the manuscript might be improved if the final section of the results was de-emphasized, e.g. by mentioning it briefly in the manuscript and moving the data & description to supplementary information.

PLOS authors have the option to publish the peer review history of their article (what does this mean?). If published, this will include your full peer review and any attached files.

Reviewer #1: No

Reviewer #2: No

Reviewer #3: No
---

## [Decision Letter · Decision Letter 1]

19 Nov 2021

Dear Dr. Seder,

We are pleased to inform you that your manuscript 'Protective effects of combining monoclonal antibodies and vaccines against the Plasmodium falciparum circumsporozoite protein' has been provisionally accepted for publication in PLOS Pathogens.

Best regards,

James G. Beeson, MBBS, PhD

Guest Editor

PLOS Pathogens

James Kazura

Section Editor

PLOS Pathogens

Kasturi Haldar

Editor-in-Chief

PLOS Pathogens

orcid.org/0000-0001-5065-158X

Michael Malim

Editor-in-Chief

PLOS Pathogens

orcid.org/0000-0002-7699-2064

Reviewer Comments (if any, and for reference):

Reviewer's Responses to Questions

**Part I - Summary**

Reviewer #2: (No Response)

Reviewer #3: The authors have done an excellent job of revising the manuscript in response to the reviewers' feedback. I think the revised version is significantly improved and ready for publication.

**Part II – Major Issues: Key Experiments Required for Acceptance**

Reviewer #2: Authors addressed concerns satisfactorily

Reviewer #3: None

**Part III – Minor Issues: Editorial and Data Presentation Modifications**

Reviewer #2: (No Response)

Reviewer #3: None

PLOS authors have the option to publish the peer review history of their article (what does this mean?). If published, this will include your full peer review and any attached files.

Reviewer #2: No

Reviewer #3: **Yes: **Alexander Douglas

---

## [Editor Report · Acceptance letter]

2 Dec 2021

Dear Dr. Seder,

We are delighted to inform you that your manuscript, "Protective effects of combining monoclonal antibodies and vaccines against the Plasmodium falciparum circumsporozoite protein," has been formally accepted for publication in PLOS Pathogens.

Best regards,

Kasturi Haldar

Editor-in-Chief

PLOS Pathogens

orcid.org/0000-0001-5065-158X

Michael Malim

Editor-in-Chief

PLOS Pathogens

orcid.org/0000-0002-7699-2064